# LawShift: Benchmarking Legal Judgment Prediction Under Statute Shifts

**Zhuo Han**[1]*, **Yi Yang**[1]*, **Yi Feng**[1]†
**Wanhong Huang**[1], **Xuxing Ding**[1], **Chuanyi Li**[1], **Jidong Ge**[1], **Vincent Ng**[2]
[1]State Key Laboratory for Novel Software Technology, Nanjing University, China
[2]Human Language Technology Research Institute, University of Texas at Dallas, USA
`fy@nju.edu.cn`

## Abstract

Legal Judgment Prediction (LJP) seeks to predict case outcomes given available case information, offering practical value for both legal professionals and laypersons. However, a key limitation of existing LJP models is their limited adaptability to statutory revisions. Current SOTA models are neither designed nor evaluated for statutory revisions. To bridge this gap, we introduce LawShift, a benchmark dataset for evaluating LJP under statutory revisions. Covering 31 fine-grained change types, LawShift enables systematic assessment of SOTA models' ability to handle legal changes. We evaluate five representative SOTA models on LawShift, uncovering significant limitations in their response to legal updates. Our findings show that model architecture plays a critical role in adaptability, offering actionable insights and guiding future research on LJP in dynamic legal contexts.

## 1 Introduction

Legal Judgment Prediction (LJP) aims to predict judicial outcomes based on case information such as facts and claims [1, 2], and has been studied across multiple legal jurisdictions. In the U.S. and India, LJP typically centers on predicting court decisions, i.e., whether claims will be upheld, given the case facts and arguments [3, 4]. In contrast, in other jurisdictions, LJP emphasizes aligning case facts with applicable law articles [5–11]. For instance, LJP in the EU primarily focuses on predicting applicable law articles [12–14]. Similarly, Chinese LJP targets the prediction of law articles, as well as charges, and prison terms based on case facts [15, 1, 16, 17]. LJP provides timely insights into case results, supporting legal professionals and laypersons in decision-making while minimizing time consumption and the need for legal consultations.

A significant challenge in LJP is the dynamic nature of legal systems, where ongoing statutory revisions in both common and civil law continually reshape legal norms to align with evolving societal demands [18–20]. Table 1 illustrates how statutory revisions can alter case outcomes. In the example, the defendant was acquitted under the prior statute despite discharging radioactive substances into public waters; after the revision, the same act resulted in conviction. This highlights the significant impact that changes in law articles have on judicial decisions, affecting both individuals and society. Consequently, LJP models must swiftly adapt to updated statutes, revise their legal reasoning, and produce judgments grounded in current legal standards.

However, existing LJP research overlooks this challenge. First, LJP models are not explicitly designed to accommodate legal updates, leading to potential judgment errors. While off-the-shelf techniques, such as model fine-tuning [21] and model editing [22, 23], can incorporate updated legal knowledge,

---

*These authors contributed equally to this work.
†Corresponding author.

Table 1: An example of different judgment outcomes before and after revisions.

| |
|---|
| **Fact Description**: The defendant stole Tritium, a radioactive substance, from a local chemistry lab and deliberately introduced it into the public drinking water supply in his neighborhood, resulting in severe consequences, including ... |

| |
|---|
| **Old Article**: ... commits arson, breaches dikes, cause explosion, spreads poisonous substances shall be sentenced ... |
| **Judgment**: *Non-Violation* *(obsolete)* |

| |
|---|
| **Revised Article**: ... commits arson, breaches dikes, cause explosion, spreads poisonous or radioactive substances shall be sentenced ... |
| **Judgment**: *Violation* *(up-to-date)* |

they do not ensure consistent adherence to the latest statutes. Second, existing LJP evaluation typically relies on static, held-out datasets that treat law articles as unchanging, ignoring evolving legal norms [16, 17]. These datasets merge different versions of statutes under the same label [15], despite significant variations in the underlying judgment logic across versions. This conflation introduces considerable noise from judgment logic shifts during model training.

This paper investigates the ability of state-of-the-art (SOTA) LJP models to adapt to judgment logic shifts caused by statutory revisions, highlighting their limitations and guiding future research in dynamic legal contexts. To rigorously evaluate model performance amid such changes, we introduce LawShift, a Chinese statutory revision-oriented LJP benchmark featuring a metamorphic testing framework. Note that diverse statutory revisions, such as adding exceptions, redefining terms, or modifying conditions, differentially affect judgment logic, LawShift encompasses 31 fine-grained revision types to capture this complexity.

To evaluate whether LJP models capture shifts in legal reasoning caused by statutory revisions, we employ a metamorphic testing (MT) approach [24]. Traditional evaluations using metrics like F1 depend on ground-truth labels, which are often missing or ambiguous in the context of evolving laws—especially when newly revised statutes lack corresponding judged cases and labels. MT solve this limitation by defining metamorphic relations, which are logically grounded expectations about how model predictions should change following specific statute revisions.

We evaluate five SOTA LJP models using LawShift and find that they exhibit notable vulnerabilities under evolving legal norms. Most models struggle to align their reasoning with updated statutes, defaulting to outdated legal logic learned during training. Models equipped with explicit legal knowledge encoding mechanisms or fine-grained attention to statute semantics show slightly improved adaptability, while those primarily relying on statistical correlations between facts and labels perform significantly worse. In a nutshell, our contributions are three-fold.

First, we conduct the first evaluation of LJP models under statutory shifts by introducing a comprehensive dataset tailored to test model adaptability to statutory revisions. Covering 31 revision types, our dataset serves as a foundational resource for advancing LJP and legal reasoning in language models.

Second, we adopt a metamorphic testing strategy to assess whether model predictions align with revised legal reasoning, without relying on extensive manual annotations. This approach is broadly applicable to other legal tasks where ground-truth labels are unavailable or ambiguous [25–27].

Third, we conduct an in-depth evaluation of SOTA LJP models on our dataset, uncovering key strengths and limitations under statutory revisions. Our analysis of model mechanisms offers actionable insights for improving legal reasoning in dynamic legal contexts.

## 2 Related Works

Current SOTA methods primarily focus on improving performance on existing benchmarks (such as ECtHR and CAIL-2018) [12, 15] with static law labels, employing deep learning models to address LJP subtasks. Some approaches propose neural network-based methods that capture the inherent dependencies among LJP subtasks, thereby enhancing model performance [1, 28]. Others make predictions by reasoning over legal knowledge expressed through the semantics of subtask labels [28, 16]. With the advancement of Pretrained Language Models (PLMs), researchers have developed PLMs specifically pretrained on extensive legal texts [29, 30]. These tailored models are designed to better handle the nuances of legal language and can be fine-tuned for specific LJP subtasks. The rise of powerful Large Language Models (LLMs) has led to their rapid adoption in LJP, with several LLM-based approaches achieving notable success [31, 32].

To the best of our knowledge, no prior work has explored LJP under statutory revisions, a crucial yet overlooked aspect for real-world applicability. Existing datasets conflate different versions of law articles under static labels, introducing legal reasoning noise during training. Moreover, current SOTA models lack specifically designed mechanisms to accommodate evolving statutes. While strategies like model fine-tuning and editing can inject updated legal knowledge, they do not ensure LJP models consistently align with revised statutes, resulting in prediction errors [33, 34]. A related work by Wang et al. [35] benchmarks LLMs' ability to handle edited legal knowledge, focusing primarily on evluating updating mechanisms. They apply various updating mechanisms (e.g., RAG [36], ROME [33], LoRA [37]) to multiple LLMs and evaluate their performance. In contrast, our work focuses on evaluating LJP models' ability to adapt to shifts in underlying judgment logic induced by statutory revisions. Our dataset encompasses 31 distinct types of statutory changes, far exceeding the three covered by prior work. Additionally, their dataset's fact descriptions are highly condensed and simplified, lacking the complexity of authentic case narratives. This oversimplification risks biasing evaluations and undermining objectivity, as LJP critically depends on nuanced factual details to align with statutory elements. Our dataset preserves the richness and structural complexity of real-world cases, enabling a more realistic and robust assessment of models' true capability in statute shift scenarios. Another related study [25] examines LJP models' capacity to extract key elements and ensure fairness across sensitive attributes like race. By contrast, our work probes models' sensitivity and adaptability to judgment logic shifts induced by statutory revisions, further analyzing how architectural differences affect their legal reasoning adaptability.

# 3 LawShift

This study targets the Chinese Criminal Code [3]. We detail the annotation process of LawShift.

## 3.1 Identifying Statutory Revision Types

A law article consists of two core components: the *constitutive element*, which specifies the factual conditions required for the law to apply, and the *legal consequence*, which defines the resulting outcomes or penalties when those conditions are met. In practice, statutory revisions modify these components through diverse changes. Note that real-world revisions are often coarse-grained, with multiple component changes occurring simultaneously. However, our aim is to evaluate LJP models on as fine-grained revision types as possible.

To systematically identify fine-grained statutory revisions, we first introduce a **Law Article Template** outlining key legal article components may change. Next, we define a comprehensive set of **Amendment Dimensions** that capture all possible ways these components can be changed (i.e., how components change). We analyze all 12 historical editions of the Chinese Criminal Code Amendments to derive empirically grounded revision types, which define the final statutory revision taxonomy of LawShift.

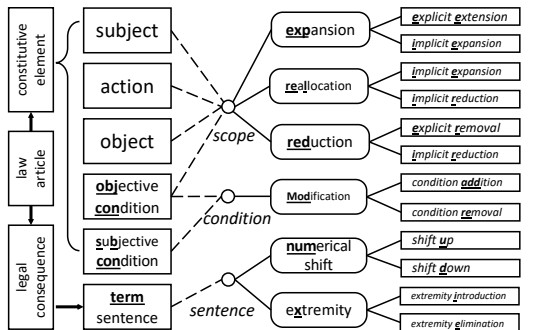

Figure 1: Law article template (left) and amendment dimensions (right).

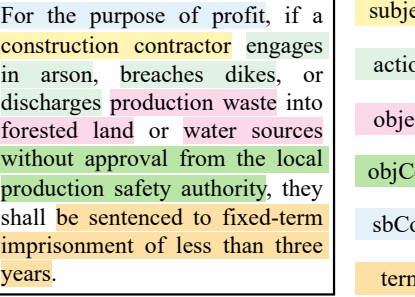

Figure 2: A fictional law article crafted to includ all components within a single example.

[3]We emphasize that while our dataset is constructed from cases and laws in China, the methodology is applicable across other legal jurisdictions. Further discussions of generalization can be found in Appendix B

Table 2: Details of the three scope changes. $S$ and $S'$ denote the scopes before and after revision

| TYPE | DEFINITION | DESCRIPTION | EXAMPLE |
|---|---|---|---|
| expansion | $S \subsetneq S'$ | the scope of a component is broadened | *adults → anyone above 16* |
| reallocation | $S \cap S' \neq \emptyset$ and $S \triangle S' \neq \emptyset$ | the new scope added and removed certain entities or actions previously covered | *state-owned enterprise employees → executives of any corporation* |
| reduction | $S' \subsetneq S$ | the scope of a component is narrowed | *consume and traffic drugs → consume drugs* |

**Law article template** Under guidance from legal experts, we decompose law articles into multiple sub-components, as illustrated in Figure 1, with the *constitutive elements* further divided into five sub-components: (1) **_subject_**, which defines the scope of the individuals or groups responsible for the crime; (2) **_action_**, which specifies the scope of the criminal actions by the *subject*; (3) **_object_**, which identifies the scope of entities involved in or affected by the *action*; (4) **_obj_**ective **_con_**dition, which outlines situational constraints under which the action becomes criminal; (5) **_su_**bject **_con_**dition, which captures the required intent or mental state of the *subject*. The *legal consequences* is represented by a single sub-component: the *term sentence*[4]. Figure 2 presents an example of law articles.

**Amendment dimensions** We identify possible changes and organize them into a hierarchical structure, as illustrated in Figure 1.

There are three primary revision types: changes in *scope* definitions, *conditions*, and *sentences*. The *scope* definition changes involve components that specify the defined boundaries of entities or actions subject to regulation under the corresponding articles. This type of change involves four sub-components, i.e., *subject*, *actions*, *object*, and *objective condition*. Recall the example in Table 1, after the *scope* of the *object* changs from *poisonous substances* to *poisonous or radioactive substances*, the case's outcome shifted from *Non-Violation* to *Violation*. The *scope* can be further classified into three types: expansion, reduction, and reallocation, as shown in Table 2. We can further categorize expansion, reallocation and reduction changes into two subtypes, i.e. *explicit extension/removal* and *implicit rephrasing*. Explicit scope changes occur through enumerative or parallel modifications, whereas implicit rephrased scope changes involve a semantic shift marked by a complete rewording of the component's definition, as illustrated in Table 3.

Similarly, the *condition* changes involve *subjective condition* and *objective condition*. These revisions typically manifest either through the addition of a condition or the removal of an existing one. The addition of a condition may introduce new criteria or prerequisites that must be met for the law to apply, potentially making the legal requirements more strict.

The *sentence* change involves the *term sentence* component in an article's *legal consequences*. This type of change can be further divided into two categories, namely *numerical shift* and *extremity*. *Numerical shift* refers to adjustments in penalty ranges, specifically *shift up* and *shift down*. *Extremity* represents non-numerical modifications, including the *introduction* or *elimination* of discrete sentencing terms, such as *life imprisonment* or the *death penalty*.

Table 3: Examples of explicit and implicit scope changes

| TYPE | EXAMPLE |
|---|---|
| *explicit extension* | *poisonous substances → poisonous or radioactive substances* |
| *implicit expansion* | *adults → anyone above 16* |
| *explicit removal* | *consume and traffic drugs → consume drugs* |
| *implicit reduction* | *corporation employees → corporation executives* |

Table 4: Examples of metamorphic testing in LawShift.

| TYPE | LAW RVSN. | FACT ALT. | EXPT |
|---|---|---|---|
| **T1.1** | *Article 271*: ... personnel of **state-owned companies**, ... → ... personnel of **private enterprises, state-owned companies**, ... | ... the defendant while serving as person in charge of company A, which is a **state-owned company**, ... → ... the defendant while serving ... a **private enterprise** B ... | V |
| **T6.1** | *Article 232*: ... the sentence shall be **3 to 10 years** imprisonment. → ... the sentence shall be **more than 10 years** imprisonment. | N/A | T ↑ |

---

[4]Other penalties, such as fines, are inconsistently specified across statutes and often ignored by existing LJP models, thus excluded from our study.

Table 5: Revision types, corresponding testing methods, and the number of test instances in LawShift.

| TYPE | BRIEF DESCRIPTION | TASK | EXPT. | # |
|---|---|---|---|---|
| **T1.1**: subject-exp-ee | identify subject in the explicitly extended scope | Article | V | 261 |
| **T1.2**: subject-exp-ie | identify subjects in the implicitly expanded scope | Article | V | 261 |
| **T1.3**: subject-rel-ie | identify subjects in the expanded scope of the reallocation | Article | V | 261 |
| **T1.4**: subject-rel-ir | identify subjects in the reduced scope of the reallocation | Article | NV | 261 |
| **T1.5**: subject-red-er | identify subjects in the explicitly removed scope | Article | NV | 261 |
| **T1.6**: subject-red-ir | identify subjects in the implicitly reduced scope | Article | NV | 261 |
| **T2.1**: action-exp-ee | identify actions falls in the explicitly extended scope | Article | V | 102 |
| **T2.2**: action-exp-ie | identify actions falls in the implicitly expanded scope | Article | V | 102 |
| **T2.3**: action-rel-ie | identify actions falls in the expanded scope of the reallocation | Article | V | 102 |
| **T2.4**: action-rel-ir | identify actions falls in the reduced scope of the reallocation | Article | NV | 102 |
| **T2.5**: action-red-er | identify actions falls in the explicitly removed scope | Article | NV | 102 |
| **T2.6**: action-red-ir | identify actions falls in the implicitly reduced scope | Article | NV | 102 |
| **T3.1**: object-exp-ee | identify objects in the explicitly extended scope | Article | V | 459 |
| **T3.2**: object-exp-ie | identify objects in the implicitly expanded scope | Article | V | 459 |
| **T3.3**: object-rel-ie | identify objects in the expanded scope of the reallocation | Article | V | 459 |
| **T3.4**: object-rel-ir | identify objects in the reduced scope of the reallocation | Article | NV | 459 |
| **T3.5**: object-red-er | identify objects in the explicitly removed scope | Article | NV | 459 |
| **T3.6**: object-red-ir | identify objects in the implicitly reduced scope | Article | NV | 459 |
| **T4.1**: objCon-exp-ee | identify entities in the explicitly extended scope and determine non-compliance | Article | NV | 150 |
| **T4.2**: objCon-exp-ie | identify entities in the implicitly expanded scope and determine non-compliance | Article | NV | 150 |
| **T4.3**: objCon-rel-ie | identify entities in the expanded scope of the reallocation and determine non-compliance | Article | NV | 150 |
| **T4.4**: objCon-rel-ir | identify entities in the reduced scope of the reallocation and determine compliance | Article | V | 150 |
| **T4.5**: objCon-red-ir | identify entities in the implicitly reduced scope and determine compliance | Article | V | 150 |
| **T4.6**: objCon-mod-add | determine non-compliance with the added objective condition | Article | NV | 409 |
| **T4.7**: objCon-mod-re | determine compliance with the removed objective condition | Article | V | 149 |
| **T5.1**: sbCon-mod-add | determine non-compliance with the added subjective condition | Article | NV | 470 |
| **T5.2**: sbCon-mod-re | determine compliance with the removed subjective condition | Article | V | 115 |
| **T6.1**: term-num-up | sentence longer terms | Term | T ↑ | 215 |
| **T6.2**: term-num-dn | sentence shorter terms | Term | T ↓ | 215 |
| **T6.3**: term-x-in | sentence extreme terms | Term | XT | 215 |
| **T6.4**: term-x-el | no longer sentence extreme terms | Term | NX | 99 |

**Determining the final revision types** Following these steps, we define all potential statutory revision types as component-amendment pairs (Figure 1). However, not all theoretically possible types occur in practice. After reviewing all 12 historical editions of the Chinese Criminal Code Amendments, LawShift incorporates only those revision types empirically observed in real-world. The finalized types and their detailed descriptions are provided in Table 5.

## 3.2 Testing Methodology

In LJP, ground-truth labels are often unavailable or ambiguous under evolving statutes as newly revised statutes may lack corresponding real-world cases and no golden label exist. While expert annotation is an option, these labels are annotator-generated rather than court-issued, raising concerns about judicial authenticity and undermining the objectivity and fairness of evaluation. Thus, traditional evaluation methods (e.g.,computing the F1 score for law article prediction task) are not suitable for statutory revisions, as they rely on ground-truth labels that may be missing under shifting statutes. Metamorphic testing allows for the testing of situations where it may not be feasible to define ground-truth labels [24]. Specifically, metamorphic testing assesses whether LJP models produce expected outputs consistent with the judgment logic introduced by statutory revisions. We employ two kinds of metamorphic testing. One is comparing the outcomes of LJP models before and after revisions (**T6.1**-**T6.4**). As shown in Table 4, for revision type **T6.1** (i.e., changes to statutory sentencing ranges), we evaluate whether LJP models adjust sentencing predictions in line with updated judgment logic. For instance, if Article 232's range shifts from "3–10 years" to "more than 10 years", a model passess the test if cases previously predicted within 3–10 years now yield predictions exceeding 10 years. Here, ground-truth labels are unnecessary—we only assess whether the model outputs a value exceeding 10 years. The other kind of metamorphic testing constructs minimal test instances targeting a single capability, e.g., recognizing a legal condition change (**T1.1**-**T5.2**). For example, in Article 271 in Table 4, originally applied only to state-owned enterprise personnel but was extended to include private enterprise individuals. We create a minimal test by modifying real cases judged under the pre-revision statute, i.e., replacing "state-owned enterprise" with "private enterprise" while

keeping other facts unchanged. If If the LJP model predicts a violation, it reveals that the model internalizes logic changes (e.g., statutory revisions). This strategy avoids crafting entirely new cases and relies on targeted text edits to reflect statutory changes [5].

## 3.3 Generating Test Instances

Next, we detail how we generate test instances.

**Collecting original cases**   We construct LawShift test instances based on real-world judicial documents from China Judgments Online (CJO)[6]. Specifically, we select cases issued between 2017/11/4 and 2021/2/28 as these cases share the same version of laws and judged by the same judgment logic. By controlling the time span, we ensure that the original judgments are consistent with the legal framework in effect at the time, allowing us to attribute changes in model behavior solely to the introduced statutory revisions and avoid interference from other legal contexts or temporal discrepancies [38] [7].

We then construct LawShift using the original legal cases collected above. Each test instance consists of three elements: a pair of law articles (before and after the revision), a corresponding pair of fact descriptions (from the original case and its edited version), and an expected outcome aligned with the revised law article. During the annotation process, we engaged one senior annotator (a law school professor) and two annotators (law school PhD candidates) to ensure the legal expertise of LawShift.

**Revising law articles**   Next, we introduce revisons to law articles. For each revision type in Table 5, annotators collaboratively identify suitable law articles and manually construct corresponding revised versions that reflect the intended changes. For instance, for revision type **T4.7** (removal of an *objective condition*), only law articles containing this component are eligible for revision. In practice, one annotator independently drafts each revised law article based on the target revision type, which is then reviewed by a second annotator for legal validity, coherence, and alignment. Conflicts are resolved through discussions led by a senior annotator, who iteratively refines the drafts to ensure consensus.[8]

To ensure fair comparison of LJP model performance across different revision types affecting the same component, some revision types share identical selected law articles and original cases. For example, **T6.1** and **T6.2**, which involve revisions to sentencing terms, share the same set of selected law articles. This design ensures that, for each component, the impact of its different revison types can be compared fairly without being affected by doctrinal variations, or data distribution differences.

**Editing facts**   For each revision type, we retrieve original cases citing the corresponding law article from the previously selected set. We then edit the retrieved case facts, when necessary, to align with the corresponding revised law article. For instance, fact editing is required when testing revision types involving *constitutive elements*, as we adopt a minimal test strategy (Section 3.2). For example, if a statute's *subject* expands from "poisonous substances" to "poisonous or radioactive substances", we edit the original case facts to replace the poisonous substance with a radioactive one. This minimal change isolates the effect of the revision, enabling us to assess whether the model correctly recognizes the inclusion of radioactive substances under the revised statute. As for **T6.1-T6.4**, we use the original facts as we adopt the comparison strategy (Section 3.2).

There are three approaches we've used to revise the case facts: (1) perturbing existing cases, (2) employing LLMs to synthesize new facts, and (3) using real case facts.

For revisions targeting *subject*, *object*, *objective condition*, and *subjective condition* ( **T1.1–1.6**, **T3.1–3.6**, **T4.1–4.7**, **T5.1–5.2**), we apply the perturbation strategy. This involves (1) identifying relevant fact spans using regular expressions, and (2) inserting or replacing them with phrases from predefined sets curated by our three expert annotators during the law revision process. For instance, when editing facts for **T5.1**, where the *subjective condition* may be revised to "out of revenge or to vent frustration", we build a phrase list describing alternative motives unrelated to this condition to prompt non-violation predictions. Using regular expressions, we locate the defendant span in

---

[5]See Table 13 (Appendix H) for details of revision types with their testing methods, examples.
[6]http://wenshu.court.gov.cn/
[7]See Appendix C for pre-processing details.
[8]The annotators were paid an hourly wage of 200 RMB.

the case facts and prepend a randomly selected phrase from the list. This perturbation approach ensures minimal, targeted edits specific to the revision type, while preserving the overall coherence and integrity of the case facts.

LLMs are employed to edit case facts for revision types **T2.1**–**T2.6**, which involve changes to *action scopes*. Unlike entities—typically isolated noun phrases—actions are embedded within complex narrative structures, and revisions to their scope often alter the semantics of the entire case. Such complexity makes simple perturbation insufficient. To address this, we leverage GPT-4o [39] to generate coherent, legally sound case facts that reflect the revised statutory definitions. GPT-4o is guided via one-shot prompting, using exemplar pairs crafted by annotators to ensure alignment with the revision intent and judicial style. Specifically, two annotators independently draft one exemplar for the revised law articles of each revision type. Each exemplar consists of a pair of case facts (before-and-after revision) aligned with the corresponding revised law article. Subsequently, the senior annotator reviews and refines these exemplars when necessary [9].

Revision types **T6.1**–**T6.4**, which involve changes to *legal consequences*, do not require modification of case facts. As illustrated in Table 4, these revisions can be evaluated by comparing a model's sentencing predictions on unchanged facts under the original and revised statutes. Thus, the original case facts are directly reused as the edited ones [10].

**Expected Output** An LJP model is considered to pass a test instance in LawShift if it generates the expected output given the edited fact. Expected outputs for each revision type are defined in Table 5. For example, in **T1.1** (subject extension), if a model correctly identifies a subject within the newly extended scope and predicts **Violation** (V), it passes. Other expected outputs include **Non-Violation** (NV), **Increased Term** (T ↑), **Decreased Term** (T ↓), **Extreme Term** (XT), and **Non-Extreme Term** (NX). Different LJP tasks are used depending on the nature of each revision type.

**LawShift statistics** LawShift encompasses a total of 7,569 test instances[11]. The statistics of test instances for each revision type is shown in Table 5.

# 4 Experiments

## 4.1 Evaluation Setup

**Base dataset** Before evaluating LJP models on LawShift, we construct a base dataset for model training. This dataset, like LawShift, is sourced from China Judgment Online and processed using the same pipeline (see Appendix C). To ensure consistent judgment logic, we select cases from the same time span as LawShift. The final dataset includes 253,936 training and 63,484 validation cases.

**Baselines and metrics** We evaluate nine SOTA (TopJudge [1], D-LADAN [16], NeurJudge [40], Lawformer [30], Qwen2.5-7B, Llama3-Instruct-8B, ChatGLM3-6B, Llama3-8B-Law, and Law-model-7B) on LawShift and report the *Pass Rate* for each revision type—where a case is considered a *pass* if the model generates the expected outcome given the edited fact. Each baseline represents a distinct family of LJP modeling approaches, some of which are inherently capable of adapting to statutory revisions. For example, NeurJudge and Lawformer include mechanisms for matching with law article texts; during evaluation, we provide them with the revised statutes as input. Similarly, LLM-based models are evaluated using prompts that include the revised law text to assess their ability to adapt to changes in judgment logic. We conduct two evaluations (before and after revision) per revision type on LawShift: (1) assessing model performance on original judgments using original facts, and (2) assessing performance on expected outcomes using the edited facts. This can help us fairly compare LJP models' performance across different components. Specifically, we compute normalized relative pass rate shift $\mathcal{R}_c$ (of component $c$), computed as $\mathcal{R}_c = \sum_{i=1}^{n_c} 2 * (r_{f_i} - r_{\text{ori}_i})/(r_{f_i} + r_{\text{ori}_i})$ where $r_{f_i}$ denotes the pass rate under revision and $r_{\text{ori}_i}$ represents the pass rate before revision for the $i$-th revision type of component $c$. $n_c$ is the number of different revision types of component $c$. Data

---

[9]See Appendix H.2 for prompt and exemplar details.

[10]Examples of how we revise law articles and corresponding facts can be found in Appendix H.

[11]Our dataset and code are available on `https://huggingface.co/datasets/triangularPeach/LawShift/tree/main` and `https://github.com/triangularPeach/LawShift`.

distributions vary across components, e.g., the original cases for **T1.1** and **T2.1** differ. $\mathcal{R}_c$ mitigates these distributional differences, enabling fair comparisons between components (e.g., *subject* vs. *action*). To compare models across different revision types of the same component, we analyze $r_{f_i}$, since these revision types share the same selected law articles and original cases (as mentioned in Section 3.3). Details on baselines, metrics, and experimental settings are provided in Appendix D.

## 4.2 Performance Analysis

We report experimental results in Figure 3 (See Table 11 in Appendix for detailed values).

**Overall results**   In Figure 3, the revision types indicated by the gray background (e.g., **T1.2**, **T4.5**) represent types where predictions are expected to remain consistent, whereas those highlighted with a beige background (e.g., **T1.4** and **T5.1**) correspond to revisions expected to lead to changes in predictions. We can find that, most neural-based LJP models perform reliably when predictions are expected to remain consistent but struggle to adjust when legal revisions require different outcomes, highlighting their limited adaptability to statutory changes, while LLM solutions of LJP exhibit higher level of performance on legal revisions require altered sentence.

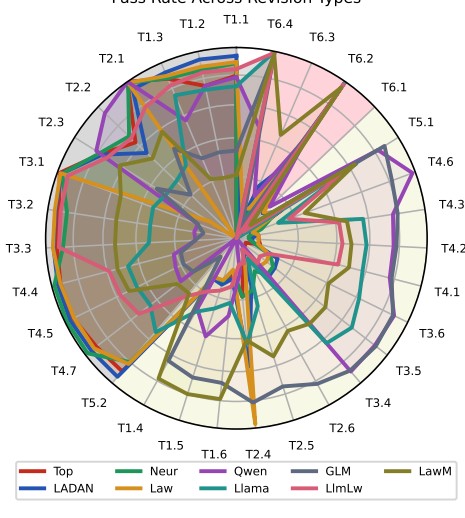

Figure 3: Pass rate across revision types.

Notably, **T2.2** and **T2.3** are often misclassified under an unrelated law article, specifically, *drug possession*, which is a pattern not seen before revision. This may be due to revised cases including phrases like *harboring individuals for drug transactions*, which semantically overlap with possession offenses. The pattern suggests Lawformer relies more on surface-level keywords than on accurate legal interpretation in context.

Revision types related to prison terms are highlighted with a pink background. We observe that models achieves higher pass rates on revisions that reduce sentence severity. Specifically, models perform better on **T6.2** than **T6.1** and on **T6.4** than **T6.3**. This trend indicates a bias toward lighter penalties, possibly due to the training data's skew toward lenient outcomes.

**Normalized Relative Pass Rate Shift**   We further examine model sensitivity to changes to different components using the metric $\mathcal{R}_c$. In Figure 4, all neural-based models demonstrate similar level of performance on *subject* and *object* revisions, suggesting these components require similar reasoning process. In contrast, neural-based models perform relatively worse on objective condition revisions, which demand additional reasoning over scope and conditional logic, while performing better on subjective condition revisions that mainly involve adding or removing single conditions.

General purpose LLMs show clear variation in their sensitivity across reasoning components. As shown in Figure 4, LLMs exhibit larger shifts on *object* and *subjective condition* components, indicating that revisions involving semantic role changes or the addition and removal of qualifying conditions are the most likely to affect their predictions. These two components require the model to reinterpret local contextual dependencies while preserving global consistency, which increases the reasoning complexity compared with static entities or structural terms. In contrast, smaller shifts are observed for *subject* and *objective condition* components, suggesting that identifying core entities or scope boundaries is less challenging for models trained on broad text corpora.

Across all models, *term* revisions and *action* revisions remain the most challenging, with traditional baselines showing the lowest normalized pass rate and only marginal improvement from large models. For *term* revisions, the results imply that sentence-term reasoning still poses a persistent limitation even for architectures with large-scale pretraining, as it requires quantitative alignment between case outcomes and statutory boundaries rather than purely linguistic understanding. *Action* revisions also pose a greatest challenge for models, likely because understanding revisions to cases' actions requires deeper contextual comprehension [12].

---

[12]See Appendix F for details on *action* revision error analysis.

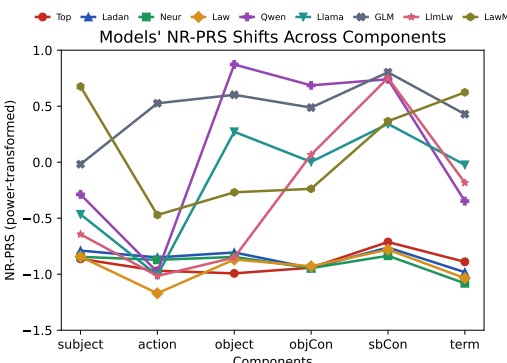

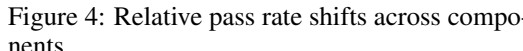

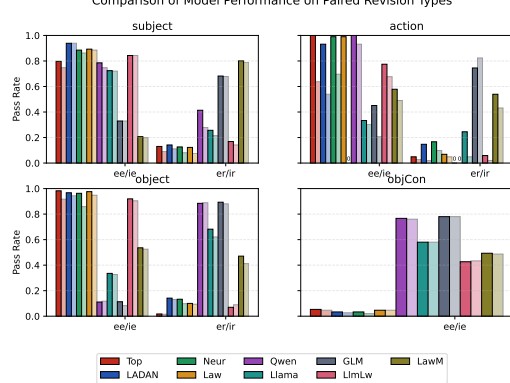

Figure 4: Relative pass rate shifts across components.

Figure 5: Pass rate differences between explicit and implicit changes. Lighter shades represent implicit changes (ie/ir).

**Pass Rates between Formulations**    As described in Section 3.1, we define two formulation types for scope revisions: *explicit extension/removal* and *implicit rephrasing*. Figure 5 presents model pass rates on revision types involving such distinctions, specifically **T1.1-1.2; 1.5-1.6**, **T2.1-2.2; 2.5-2.6**, **T3.1-3.2; 3.5-3.6**, and **T4.1-4.2**. It is evident that all models perform better on explicit changes than on their implicit counterparts. This supports our hypothesis that *implicit rephrasing* poses greater challenges, likely due to its higher demand for semantic understanding of legal language.

## 4.3    Discussions

Among the nine models, TopJudge performs worst due to its lack of law article awareness, underscoring that incorporating legal text boosts adaptability. Models like D-LADAN and NeurJudge, which explicitly reference law articles for legal differentiation or key fact extraction, show better robustness but still struggle with revisions requiring deeper semantics understanding, such as *implicit rephrasing* or *action* revisions. Meanwhile, the general-domain LLMs adapt flexibly via in-context learning, but their outputs are generally inconsistent due to prompt sensitivity and pretraining bias. For general purpose LLMs, their pretraining enables stronger generalization to context-dependent changes, particularly when modifications involve nominal or semantic role adjustments, but also makes them more sensitive to surface-level variations in conditional structures. Among them, Chat-GLM maintains relatively stable shifts across all components, indicating balanced sensitivity to both entity and logical changes, whereas Llama shows more fluctuation between object and term components, implying partial over-reliance on lexical co-occurrence learned from open-domain text. On the other hand, law-oriented models demonstrate a distinct bias towards condition-level reasoning. They perform consistently on *subject* and *object* components but show higher positive shifts on *objective* and *subjective condition* components, suggesting an increased capacity to encode legal relational dependencies once pretraining includes domain statutes or fine-tuning on legal corpora.

To improve LJP models under statute shifts, future research should focus on several directions. First, **better utilization of law content** is needed. Models should move beyond keyword matching and learn context-aware representations that capture legal hierarchies and fact-law matching. Second, **handling implicit rephrasings and action changes** remains a key challenges. Current models struggle with semantic abstraction, calling for contrastive learning and event-level representations tailored to legal text beyond surface-level understanding. Third, **balancing stability and adaptability** is difficult. LLMs like Qwen adapt well in context but lack consistency. Hybrid methods such as adaptive RAG may help by grounding predictions in dynamic legal context while maintaining structured reasoning. Finally, **temporal legal learning is essential**. Models should support timestamp-aware pretraining and continual learning to track legal amendments without full retraining. Moreover, considering that in real-world settings, post-amendment data is scarce, making full retraining impractical. This calls for training frameworks that treat time as part of supervision, guiding models to align predictions with the valid law at the relevant time, even under low-resource conditions.

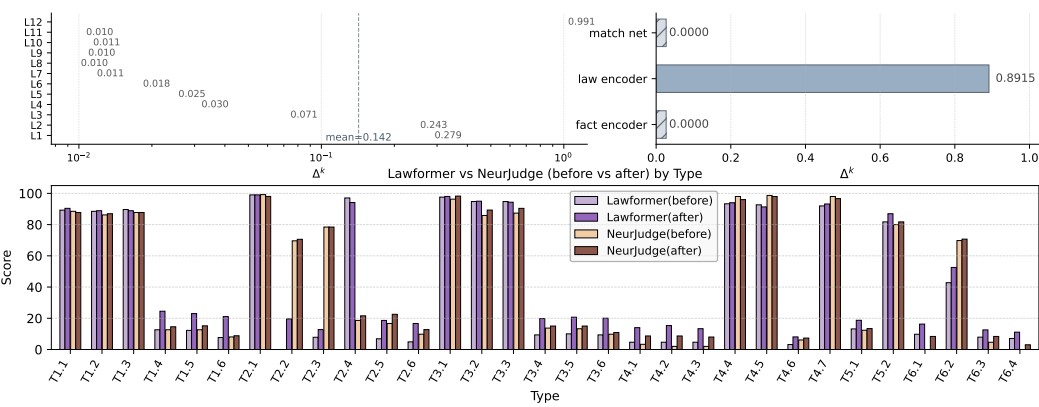

Figure 6: (top) Layer-wise and module-wise hidden-state shifts ($\Delta^k$) with updated statutes of Lawformer and NeurJudge, respectively. (bottom) Post-editing performance across revision types after applying localized model weight updates.

## 4.4 Broader Evaluation: Knowledge Tracing and Editing

To further explore how model responds to statutory revisions and whether post-hoc updates can effectively adapt models, we conduct broader evaluation experiments using knowledge tracing [41, 42] and model editing [33] on Lawformer and NeurJudge.

**Knowledge Tracing** We trace hidden representations of both models before and after statutory revisions to reveal how internal layers encode legal changes. Figure 6 (top-left) shows that Lawformer exhibits the largest change at its final encoder layer (Layer 12), indicating that statute revisions mainly influence high-level semantic representations. In contrast, NeurJudge displays concentrated variation in its law-encoder component, while its fact encoder and matching module remain nearly stable, as illustrated in Figure 6 (top-right). These findings identify the most revision-sensitive layers, providing concrete anchor points for subsequent lightweight editing [13].

**Knowledge Editing** Building on the tracing analysis, we took inspirations from the model editing method ROME [33] and applied localized rank-one updates to the most sensitive position in the respective models. As illustrated in Figure 6 (bottom), both backbones benefit from editing, as both models show improvements across most revision types. However, the gains are still rather small, and the results suggest that localized parameter updates can alleviate certain surface-level mismatches but fall short of capturing deeper judgment logic shifts. More principled approaches, such as structure-aware continual learning or representation-level adaptation, are still required to achieve robust responsiveness under statutory revisions.

## 5 Conclusion

We propose LawShift, a benchmark for evaluating LJP models under statutory revisions, covering 31 revision types across 6 categories, each aligned with specific expected model behaviors. We assess five representative SOTA LJP models, revealing their strengths, limitations, and design implications for improving adaptability to legal changes.

**Limitations** Despite LawShift's jurisdictional and linguistic constraints, its core testing methodology (decomposing legal revisions, aligning them with model behavior, and evaluating reasoning consistency) is broadly generalizable across languages and legal systems. We evaluate baselines with limited revision-handling capabilities, e.g., law-aware attention or prompt injection. Nonetheless, our focus is identifying how models fail under statute shifts, rather than solving the update process.

---

[13]Details of these experiments are shown in Appendix G.1.

## Acknowledgments

We thank the reviewers for their valuable comments on an earlier draft of this paper. This work was supported by National Natural Science Foundation of China (No. 62406139), State Key Laboratory for Novel Software Technology at Nanjing University (KFKT2025A15, ZZKT2025B14, KFKT2024A07, ZZKT2024B02).

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

## A    Ethical Considerations

The original case data used during LawShift construction are all from publicly available resources. When using non-locally deployed large language models, all case data inputs were properly anonymized (e.g. defendant names, location, etc.) to mitigate the risk of privacy leaks. Our motivation is to evaluate the performance of LJP models under statutory revisions, with the aim of helping researchers in developing more robust LJP systems. We do not intend to assert that LJP models are capable of adapting to legal changes to the extent that they can be deployed independently without human verification.

## B    Generalization and Scalability of LawShift

### B.1    Generalization

For the design and construction of LawShift, the method of statute decomposition and revision type definition is widely generalizable. Regardless of legal system, codified rules can be represented as propositional statements or structured clauses, which our decomposition strategy is built to handle. For example, the UK's Sentencing Council specifies key elements and penalties for Attempted Murder [14], noting "... the offender had an intention to kill; accordingly and offender convicted of this offence will have demonstrated a high level of culpability ..." with "maximum: life imprisonment; offence range: 3-40 years' custody." Using our approach (see Figure 1), this statute can be decomposed into core component: subject (offender), action (kill), subject condition (intention), and legal consequences (sentence range: 3-40 years). Although our revision taxonomy was developed from the Chinese Criminal Code, many types like T6.1 (term-num-up, referring to Table 5) and T2.1 (action-exp-ee) apply directly to such UK statutes.

Second, our metamorphic testing approach for assessing model adaptability to legal revisions is broadly applicable. For instance, in the US, complexities arise when both statutes and precedents change. The 2022 Supreme Court decision in Dobbs v. Jackson Women's Health Organization [43] overturned the 1973 Roe v. Wade [44] precedent that recognized abortion as a constitutional right, shifting regulatory authority back to the states. While Roe invalidated state-level abortion bans, Dobbs allows states to criminalize abortion. Thus, a Texas doctor performing abortions in 2021 acted lawfully, but identical conduct in 2023 may lead to prosecution, reflecting the 2022 state legislation banning abortion.

We can design metamorphic test cases that capture the interplay of statutory and precedent changes (as shown in Table 6). By presenting identical facts with variations in time, region, or precedent, we

Table 6: Judgment shift examples after statute/precedent change in the US.

| TEST ID | YEAR | REGION | FACT | APPLICABLE STATUTE/PRECEDENT | EXPECTED PREDICTION | REASONING CHANGE POINT |
|---|---|---|---|---|---|---|
| 1 | 2021 | Texas | Abortion | Roe v. Wade | Not a Crime | Based on Roe, the federal constitutional right to privacy protects a woman's decision to have an abortion; states cannot impose a criminal prohibition. |
| 2 | 2023 | Texas | Abortion | State law + Dobbs | Crime | After Dobbs overturned Roe, abortion regulation authority returned to the states. Texas enacted laws banning abortion, now enforceable without Roe's limits. |
| 3 | 2021 | California | Abortion | State law + Roe | Not a Crime | While Roe established a federal protection baseline, California also had explicit state laws affirming abortion rights, reinforcing the non-criminal status. |

evaluate whether models adjust their judgments accordingly, reflecting adaptive legal reasoning. For example, in the first two abortion cases with identical facts but differing in time and precedent, the model should classify the later case as a crime and the earlier one as not a crime.

---

[14]https://sentencingcouncil.org.uk/guidelines/attempted-murder/

## B.2 Scalability

The process of the construction of the LawShift dataset is scalable and sustainable. We can leverage regex or LLMs to enable semi-/fully-automatic data annotation. For instance, when the statutory subject expands from "poisonous substances" to "poisonous or radioactive substances," an LLM can detect the original span and replace it with a sampled entity (e.g., "radium") from a predefined list. This supports factual diversity but may introduce span-matching errors. To improve reliability while limiting manual effort, we can adopt human verification or ensemble LLM voting for more robust span detection.

As described in Section 3.3, we rely on legal experts to ensure that revision type annotations are legally plausible, e.g., avoiding unrealistic edits such as revising a theft charge into a capital offense. However, to scale the annotation of revision types, we may also adopt semi-automatic or automatic methods. We can first use tools to locate the revision span (e.g., the penalty clause), and then replace or insert content from a curated set of candidates (e.g., alternative penalties for theft). Subsequently, expert review or voting-based aggregation can be used to reduce technical errors. Note human verification is applied at critical points to ensure legal fidelity, balancing cost and quality. Thus, while legal expertise is necessary, our workflow can be designed for long-term extensibility with manageable overhead.

## C Data Pre-Processing

In this section we describe in detail the data pre-processing procedure used to construct the training and validation datasets, as well as the original case pool of LawShift.

Chinese law stipulates that judicial cases must be judged according to the version of law in effect at the time the criminal behavior occurred. In cases involving multiple criminal acts, the applicable law is the one in force at the time of the *last* criminal behavior. To ensure consistency in legal standards across the dataset, we annotate each case with its corresponding law version and construct the training and validation datasets as well as the original case pool using only cases judged under the same version of the Criminal Code.

Specifically, this annotation process is performed via one-shot prompting of a general domain LLM, Qwen1.5-14B-Chat [45]. An example of prompting Qwen to extract the date of the final criminal act (translated into English) is shown in Table 7. Then, we divide all cases into groups based on

Table 7: An English-translated example of prompting Qwen to obtain a case's final crime date.

| **INSTRUCTION** |
| --- |
| You are an expert of Chinese Criminal Law. Below is a fact description of a criminal case. Please identify the criminal events involved and extract the date of the last criminal activity. If there are multiple criminal acts, extract only the date of the most recent one. The output format must be: <time>yyyy-MM-dd<eoa>. For example: <time>2001-01-09<eoa>. Please only output this data with no explanations or additional information. 
 Example: 
 **Fact**: The defendant Zhang stole a white iPhone from the victim Wang at the local mall on January 1st, 2018, ... on January 16th, Zhang broke into Li's house and stole 3,100 yuan in cash, ..., Zhang once again stole the victim Liu's wallet at the local mall on January 20th, 2018 before arrested by the police ... 
 **Output**: <time>2018-01-20<eoa>. 
 Now please process the following case fact and extract its final crime date: 
 **Fact**: On February 2, 2020, at around 1:00 am, the defendant Wu, together with and harboring Jiang, Zhang, and Yang, used the method of "blowing the bottle" to inhale the drug methamphetamine ... On February 17, 2020, at around 11:00 pm, the defendant Wu, together with and harboring Zhang, used the method of "blowing the bottle" to inhale methamphetamine ... 
 **Output**: |
| **QWEN** |
| <time>2020-02-17<eoa> |

the law version that was in effect at the time of the final criminal act in each case. The number of

cases corresponding to each law version is listed in Table 8. The time periods shown in the table are derived from the the official scripts of Criminal Code Amendments. We select the law version associated with the highest number of cases to continue to construct the subsequent datasets, i.e. the version in effect from November 4, 2017 to February 28, 2021.

Next we annotate relevant ground truth labels for charge, law article, and penalty term tasks based on the fact descriptions of selected cases. Judicial documents follow a fixed structure, typically divided into four parts: *fact description*, *court view*, *relevant laws* and *judgments*. This annotation process is carried out using precisely crafted regular expressions. Through this, we extract the *fact description* for each case, filtering out those that too short (specifically, those with fewer than 100 characters). Law article labels are extracted from the *relevant laws* section. Since law articles in these documents are cited down to the specific *paragraph*, we assign paragraph-level law article labels for each case. Meanwhile, charge labels and penalty term annotations are derived from the *judgments* section. The penalty terms are first extracted and then converted into the number of months. Labels for life imprisonment and the death penalty are also retained. An example of a pre-processed case is shown in Table 10. Finally, we divide all cases by the ratio of 16:4:5, and obtain the training and validation datasets, as well as the original case pool set, from which test instances are selected. The statistics are shown in Table 9.

Table 8: Number of cases of each law version.

| Time Period | # Cases |
|---|---|
| -99.12.24 | 1,063 |
| 99.12.25–01.08.30 | 1,307 |
| 01.08.31–01.12.28 | 328 |
| 01.12.29–02.12.27 | 1,010 |
| 02.12.28–05.02.27 | 3,298 |
| 05.02.28–06.06.28 | 3,062 |
| 06.06.29–09.02.27 | 12,121 |
| 09.02.28–11.04.30 | 23,199 |
| 11.05.01–15.10.31 | 926,094 |
| 15.11.01–17.11.03 | 696,878 |
| **17.11.04–21.02.28** | **1,187,620** |
| 21.03.01– | 11,231 |

Table 9: Statistics of used datasets.

| Dataset/Label | # |
|---|---|
| training dataset | 253,936 |
| validation dataset | 63,484 |
| case pool | 79,356 |
| charge | 56 |
| law article | 58 |
| penalty term | 12 |

Table 10: An example of pre-processed case.

| |
|---|
| **Fact Description**: The public prosecution alleges that on October 12, 2020, the defendant Qian stole an electric bicycle parked by the victim Xu near Sheshan Primary School. The bike was valued at 721 RMB. Qian was apprehended the next day and voluntarily confessed to the crime. |
| **Charge**: Theft |
| **Law Article**: Article 264, Paragraph 1 |
| **Penalty Term**: 8 months |

## D  Evaluation Setup

We test nine SOTA models with LawShift. They are:

1. TopJudge [1]: an LJP model that utilizes a directed acyclic graph to capture interdependencies among subtasks. Law Article contenst are *not* considered during prediction. TopJudge represents the family of LJP models that do not take law article contents into consideration.

2. D-LADAN [16]: an LJP model that not only employs law article contents during prediction but learns the subtle differences between similar articles in order to discriminalize confusing label semantics. D-LADAN represents the family of LJP models that utilize graph neural network to learning confusing law articles.

3. NeurJudge [28]: an LJP model that employs law article contents and use them to extract key circumstances in case facts during prediction. NeurJudge represents the family of LJP models that integrate law content to guide fact understanding and legal reasoning.

4. Lawformer [30]: a PLM pretrained on legal domain texts that can handle long legal cases. When adapted for LJP, the model was exposed with law article contents. Lawformer represents adapted PLM-based solutions.

5. Qwen2.5-7B [45]: a general domain LLM which we obtain the LJP results by directly prompting. Law article contents are provided during inference. Qwen represents LLMs in our experiments.

6. Llama3-Chinese-8B-Instruct: an instruction-tuned general-purpose LLM capable of multilingual reasoning, used zero-shot on LJP tasks. This model represents the family of open-source instruction-tuned backbones without additional legal adaptation.

7. ChatGLM3-6B: a bilingual (Chinese–English) conversational LLM optimized for reasoning and factual question answering. In our experiments, it serves as a strong generalist baseline for direct case-law prompting without domain fine-tuning.

8. Llama3-8B-Chinese-LoRA-Law: a domain-adapted version of Llama3-8B further fine-tuned via LoRA on Chinese legal corpora. This model reflects parameter-efficient adaptation of general LLMs to the legal domain.

9. Law-model-7B: a Chinese legal-specialized LLM pretrained on large-scale judicial documents and statutory texts. It represents the class of foundation models explicitly aligned with legal terminology and reasoning patterns.

The nine baselines we selected represent state-of-the-art and widely recognized approaches in legal judgment prediction (LJP), spanning traditional neural models, pre-trained legal-specific models, and LLM-based methods, three key paradigms in LJP. Notably, three of these (D-LADAN, NeurJudge, Qwen2.5, ChatGLM3, Llama3/Llama3Law) are recent models (2024–2025), ensuring our evaluation reflects both foundational and cutting-edge work. The LLM-based approaches selection was according to the performance report on legal benchmarks such as LawBench. This inclusion ensures our evaluation captures not only traditional and PLM-based methods but also the latest advances in LLM-based legal reasoning.

We conduct two evaluations (before and after revision) per revision type on LawShift: (1) assessing model performance on original judgments using original facts, and (2) assessing performance on expected outcomes using the edited facts.

For each type of revision tested, we record the **Pass Rate** for the corresponding test set. A case is deemed a *pass* if the model produces the expected result given the edited fact. The expected outcomes for each type are detailed in Table 13. All experiments are run on 4×NVIDIA RTX 3090 GPUs (24GB each). The hyper-parameters and training setting follow each model's original paper.

For each LJP model, we conducted three independent runs and reported the average results to ensure the statistical significance of the experiments.

# E    Detailed Results

The overall test results of all revision types across all models are listed in Table 11. Rows denoted with *Ori* represent the results on unaltered (original) cases. The subsequent rows starting with **T** (e.g., **T4.1**, **T4.2**, ...) correspond to different metamorphic variants derived from the same set of original cases (until the next *Ori* row). Therefore, their results are directly comparable to the preceding *Ori* row.

# F    Detailed Analyses for Performance on *Action* Components

*Actions* are typically expressed through complex event descriptions, often involving multi-step process, causative verbs, and temporal or conditional modifiers. Such event-centered constructs require understanding causality, temporal sequence, identifying cross-sentence composition, and implicit conditions, which is more demanding than recognizing the relatively static and well-defined nominal structures of subjects or objects. From a linguistics perspective, prior studies [46, 47] have shown that verbs and event structures are semantically richer and syntactically more variable than nouns, often requiring compositional interpretation across clauses. Vendler [46] argues verbs encode temporal boundaries and event dynamics, which inherently increases their interpretive complexity. Pustejovsky [47] further argues that verbs require a multi-layered semantic representation (e.g., event structure, argument structure, etc.), making the understanding of actions dependent on both context and sub-event reasoning rather than on isolated lexical cues. (i.e., what we mean by requiring deeper

Table 11: Pass rates of all models on all revision types. The results labeled as *Ori* represent the original outcomes under the unrevised laws for the corresponding revision type.

| TYPE | TOP | D-LADAN | NEUR | LAW | QWEN | LLAMA | GLM | LLMLW | LAWM |
|------|-----|---------|------|-----|------|-------|-----|-------|------|
| *Ori.1* | *90.80* | *91.57* | *91.95* | *92.34* | *59.00* | *65.90* | *47.13* | *74.71* | *19.92* |
| **T1.1** | 79.69 | 93.87 | 88.51 | 89.27 | 78.54 | 72.41 | 32.95 | 84.29 | 20.69 |
| **T1.2** | 74.71 | 93.87 | 86.21 | 88.51 | 74.71 | 72.03 | 32.95 | 84.29 | 19.92 |
| **T1.3** | 87.36 | 92.34 | 87.74 | 89.66 | 56.70 | 74.71 | 32.57 | 82.38 | 21.07 |
| **T1.4** | 13.03 | 13.41 | 12.64 | 12.64 | 29.12 | 31.03 | 64.37 | 19.54 | 78.16 |
| **T1.5** | 13.03 | 14.18 | 12.64 | 12.26 | 41.38 | 25.67 | 68.20 | 16.86 | 80.08 |
| **T1.6** | 9.20 | 11.11 | 8.05 | 7.66 | 27.97 | 21.46 | 67.82 | 14.18 | 78.93 |
| *Ori.2* | *100.0* | *100.0* | *100.0* | *100.0* | *97.06* | *80.39* | *31.37* | *91.18* | *75.49* |
| **T2.1** | 100.0 | 93.14 | 99.02 | 99.02 | 100.0 | 33.33 | 45.10 | 77.45 | 57.84 |
| **T2.2** | 63.73 | 53.92 | 69.61 | 0.00 | 93.14 | 30.39 | 20.59 | 67.65 | 49.02 |
| **T2.3** | 77.45 | 76.47 | 78.43 | 7.84 | 81.37 | 30.39 | 36.28 | 72.55 | 62.75 |
| **T2.4** | 18.63 | 56.86 | 18.63 | 97.06 | 8.82 | 43.14 | 81.37 | 5.88 | 41.18 |
| **T2.5** | 4.90 | 14.71 | 16.67 | 6.86 | 0.00 | 24.51 | 74.51 | 5.88 | 53.92 |
| **T2.6** | 2.94 | 1.96 | 9.80 | 4.90 | 0.00 | 4.90 | 82.35 | 1.96 | 43.14 |
| *Ori.3* | *98.47* | *97.82* | *96.51* | *98.69* | *10.24* | *37.69* | *14.81* | *88.02* | *59.26* |
| **T3.1** | 98.26 | 96.73 | 96.30 | 97.60 | 11.11 | 33.55 | 11.33 | 91.94 | 53.59 |
| **T3.2** | 91.72 | 94.34 | 85.84 | 94.77 | 11.76 | 32.68 | 8.28 | 90.41 | 52.51 |
| **T3.3** | 91.72 | 94.34 | 87.36 | 94.77 | 11.11 | 32.90 | 11.33 | 90.41 | 52.51 |
| **T3.4** | 1.74 | 14.81 | 13.72 | 9.37 | 88.89 | 63.18 | 88.02 | 8.06 | 47.06 |
| **T3.5** | 1.74 | 14.16 | 13.29 | 10.02 | 88.45 | 68.19 | 89.32 | 6.97 | 47.06 |
| **T3.6** | 1.53 | 13.07 | 9.80 | 9.37 | 88.89 | 62.09 | 88.02 | 8.93 | 41.18 |
| *Ori.4* | *94.67* | *94.67* | *98.00* | *92.67* | *23.33* | *40.67* | *24.00* | *40.00* | *50.67* |
| **T4.1** | 5.33 | 3.33 | 3.33 | 4.67 | 76.67 | 58.00 | 78.00 | 42.67 | 49.33 |
| **T4.2** | 4.67 | 2.67 | 2.00 | 4.67 | 76.00 | 58.00 | 78.00 | 43.33 | 48.67 |
| **T4.3** | 4.67 | 4.00 | 2.00 | 4.67 | 76.67 | 56.67 | 80.00 | 42.00 | 46.67 |
| **T4.4** | 93.33 | 95.33 | 98.00 | 93.33 | 21.33 | 46.67 | 18.00 | 58.67 | 53.33 |
| **T4.5** | 93.33 | 96.00 | 98.67 | 92.67 | 22.67 | 46.00 | 18.67 | 57.33 | 51.33 |
| *Ori.4.6* | *97.80* | *98.04* | *96.09* | *97.07* | *1.96* | *87.04* | *18.58* | *79.22* | *24.35* |
| **T4.6** | 2.69 | 2.69 | 6.11 | 3.18 | 97.78 | 12.47 | 81.91 | 17.36 | 25.22 |
| *Ori.4.7* | *95.63* | *94.63* | *97.99* | *92.62* | *23.49* | *40.94* | *24.16* | *40.27* | *75.79* |
| **T4.7** | 89.93 | 94.63 | 97.99 | 91.95 | 24.16 | 44.97 | 16.78 | 53.69 | 26.16 |
| *Ori.5.1* | *81.28* | *88.30* | *88.72* | *85.74* | *15.32* | *36.81* | *17.45* | *36.38* | *37.02* |
| **T5.1** | 16.38 | 14.89 | 12.34 | 13.19 | 82.55 | 65.11 | 88.09 | 57.45 | 64.04 |
| *Ori.5.2* | *86.96* | *93.91* | *81.74* | *81.74* | *0.00* | *55.65* | *4.35* | *9.57* | *24.35* |
| **T5.2** | 88.70 | 93.91 | 80.00 | 81.74 | 0.00 | 53.91 | 5.22 | 26.96 | 25.21 |
| *Ori.6* | *12.09* | *37.67* | *24.65* | *39.53* | *29.30* | *0.93* | *7.91* | *20.47* | *3.72* |
| **T6.1** | 0.00 | 9.30 | 0.00 | 9.77 | 13.02 | 0.00 | 3.26 | 1.40 | 3.72 |
| **T6.2** | 49.30 | 35.35 | 69.77 | 42.79 | 53.95 | 100.0 | 93.49 | 97.67 | 94.88 |
| **T6.3** | 6.05 | 17.67 | 4.65 | 7.91 | 13.02 | 0.00 | 2.33 | 0.00 | 0.47 |
| *Ori.6.4* | *77.78* | *92.93* | *90.91* | *91.92* | *53.54* | *0.00* | *10.10* | *1.01* | *2.02* |
| **T6.4** | 0.00 | 2.02 | 0.00 | 7.07 | 47.47 | 100.0 | 96.97 | 98.99 | 97.98 |

contextual understanding). In our experiments, models based on word2vec+deep learning baselines fail because their embedding representations lack the capacity to encode complex event semantics and primarily rely on co-occurrence patterns. The pretrained Lawformer model, while leveraging transformer-based contextual embeddings, is still limited by token-level attention without explicit event decomposition, bringing challenges when the action semantics are paraphrased or restructured.

# G   Details for Broader Model Evaluation

## G.1   Knowledge Tracing

We performed basic knowledge tracing [41, 42] experiments on Lawformer (Transformer-based) and NeurJudge (neural-based) and evaluated them with the action component's revision types. Our goal is to analyze how the internal representations of LJP models evolve when encountering statutory revisions, and to identify which layers or modules are most responsible for failures. We conduct layer-wise and module-level tracing on Lawformer and NeurJudge. For Lawformer, we extract hidden states from all transformer layers (L1-L12) for both original and revised clauses. For NeurJudge, as it doesn't have transformer blocks, we trace and record three vectors: (1) the final hidden state of the fact encoder, (2) the final hidden state of the law encoder, and (3) the pre-softmax output of the MatchNet. Every layer of Lawformer or every major sub-module of NeurJudge encodes a compressed summary of the statute it has read. By extracting these vectors before and after we swap in the amended clause, we obtain two snapshots of the models' internal picture of the law. Then we run the model twice. Once with the original clauses, once with the revised clause. For the two runs of experiments, we trace and compute $\Delta^k = ||v_{\text{orig}}^k - v_{\text{rev}}^k||_2$ where k indexes layers for Lawformer and the three vectors for NeurJudge. Large gap means the semantics stored in layer k changes a lot when the clause changes. That layer is therefore sensitive to the amendment. A small gap shows that the layer is largely oblivious to clause changes. By scanning through all k, we learn where in the network the new legal knowledge is (or is not) understood. The results are shown in Figure 6 (top).

For Lawformer, the strongest perturbation occurs in encoder block 12. In the variance-normalized comparison of average hidden-state distances across all transformer blocks, block 12 shows the largest mean change, indicating that this block is the point most sensitive to action-clause revisions. For NeurJudge, the largest shift appears in the law encoder final state, thus the GRU law embeddings is the bottleneck through which action revisions must be absorbed. These tracing results provide a concrete roadmap for our future work on how to adapting to revisions. For Lawformer, the pronounced sensitivity of encoder block 12 suggests that subsequent research should target this block with interventions that specialise its feed-forward space for revised action clauses. For NeurJudge, the saliency peak in the law-encoder state indicates that upcoming efforts should explore representation-level patches or lightweight gating mechanisms that operate directly on that vector. By focusing on these empirically located memory slots, future editing/updating strategies can address statutory action revisions with maximal impact and minimal collateral drift.

## G.2   Model Editing

Building on the knowledge tracing results in Section G.1, we design a targeted update mechanism to inject statutory changes into trained models without re-training from scratch. Specifically, we apply localized parameter updates to the most clause-sensitive components: encoder block 12 in Lawformer, and the law encoder's final state projection in NeurJudge. Let $\phi(x)$ denote the model's internal representation of an amended clause $x$, and let $y^*$ denote the desired updated representation that aligns with the revised legal semantics. Our goal is to perform a minimal update to model parameters $\theta$ such that $\phi(x; \theta + \Delta\theta) \approx y^*$, while leaving unrelated behaviors intact.

To achieve this, we adopt a lightweight low-rank update strategy inspired by ROME [33]. For a linear projection layer in the target module (e.g., the feed-forward block in layer 12 of Lawformer), we modify its weight matrix $W$ via a rank-one update: $W \leftarrow W + \Delta W, \quad \Delta W = uv^\top$ where $u$ and $v$ are learned directions. These directions are optimized to minimize the deviation between the edited model's output and the intended revised representation $\min_{u,v} \|\phi(x; \theta + \Delta W) - y^*\|_2^2$.

Practically, we freeze all model parameters except the selected submodule, and solve for $(u, v)$ using a least-squares objective over a small set of clause-revision pairs. For Lawformer, editing is restricted to the final feed-forward layer of encoder block 12. For NeurJudge, the law encoder's final linear

layer projecting to the match space is the only editable component. This ensures that the structural impact of the update is concentrated at the identified bottlenecks and does not affect unrelated knowledge. The test results using LawShift on the updated model is shown in Figure 6 (bottom). Despite improvements, LawShift remains challenging. Revisions relying on implicit rephrasing (T1.6, T2.6, T3.6) remain around or below 20%. Complex multi-step event reallocations are challenging for NeurJudge, and sentencing changes involving scale inversion or extreme penalties still have low success rates. These persistent errors highlight two key obstacles: many amendments shift semantics (reasoning) without explicit lexical cues, limiting token-level edits; and several revision types require richer event-centric abstraction, aligning dispersed factual mentions with revised statutes. Thus, LawShift exposes subtle semantic shifts demanding deep structural legal understanding, and while lightweight edits help, they only partially address this complexity.

## H  Details of the Revision Types

### H.1  Metamorphic Testing

Metamorphic Testing [24] is a testing method that enables systematic assessment by establishing metamorphic relations, which define how the output should change in response to specific changes of the input. Table 4 illustrates MT with examples from **T1.1** and **T6.1**.

### H.2  Fact Generation through LLMs

For the construction of revision datasets corresponding to the *action* component, we use one-shot prompting of GPT-4o to generate case facts are contextually accurate, legally consistent, and aligned with the stylistic standards of judicial fact descriptions. Table 12 is an example of prompting GPT-4o to obtain case fact for **T2.1**.

### H.3  Revision Dataset Information

The detailed information of each revision type's dataset are presented in Table 13, including the example relevant law articles before and after revision, example case facts before and after perturbation, as well as the criteria for what constitutes a ***pass***.

Table 12: An example of prompting GPT-4o to obtain case fact for **T2.1**

| INSTRUCTION |
| --- |
| **Background**: You are now required to modify a suspect's case narrative to reflect an incident of "harboring others to trade drugs." 
 **Requirements**: You must generate a modified version of the <input> based on the transformation style shown in the <reference_example>, and enclose it using the <output> tag. Do not include any procedural or case progress language, such as "under further investigation" or "subject to legal sanctions." Focus strictly on describing the criminal activity. Avoid any judgmental or conclusion-oriented legal language such as: "The above facts are clear and the evidence is sufficient" or "According to Article 354 of the Criminal Law of the People's Republic of China, the defendant should be held criminally responsible". Do not invent additional characters. The individuals involved in the narrative must only engage in the specified criminal activity. The tone of the generated narrative should match the style and tone of the example in the <reference_example>. 
 <reference_example> 
 <before_modification> ... on the afternoon of October 31, 2020, the defendant Tong instructed Zhu to purchase methamphetamine. Around 2 p.m. that same day ... he harbored Zhu for one instance of drug use. After Zhu left, Tong harbored Jin at his home for another instance of drug use. Around 7 p.m. that evening, Zhu returned to Tong's home, where Zhu and Jin used methamphetamine together. After the incident, the defendant was arrested and assisted the police in capturing drug dealer Xu ... 
 </before_modification> 
 <after_modification> 
 ... on the afternoon of October 31, 2020, the defendant Tong instructed Zhu to engage in a methamphetamine transaction. Around 2 p.m. that same day ... he harbored Zhu for one instance of drug trading. After Zhu left, Tong harbored Jin at his home for another instance of drug trading. After the incident, the defendant was arrested and assisted the police in capturing drug dealer Xu 
 ... 
 </after_modification> 
 </reference_example> 
 <input> 
 The defendant, Huang, on three separate occasions—on the evening of September 12, 2019, at noon on September 21, 2019, and in the afternoon of September 24, 2019—harbored drug users Wu and Tan at the second floor of his residence, where they used the drug methamphetamine. On the morning of September 25, 2019, police officers arrested Huang, Wu, and Tan at Huang's residence and seized a homemade drug-using device made from a plastic water bottle, a straw, tin foil, and other items. Urine tests conducted on all three individuals by the public security authorities returned positive results for methamphetamine. Forensic testing of the homemade device revealed residue containing methamphetamine. 
 </input> |

| GPT-4O |
| --- |
| <output> 
 On the evening of September 12, 2019, at noon on September 21, 2019, and in the afternoon of September 24, 2019, the defendant Huang, on three separate occasions, harbored Wu and Tan on the second floor of his residence to conduct drug transactions. On the morning of September 25, 2019, police officers arrested Huang, Wu, and Tan at Huang's residence and seized tools related to drug transactions, including a plastic water bottle, a straw, tin foil, and other items. Investigation confirmed that the tools had been used in drug transactions, and forensic analysis detected methamphetamine residue on them. 
 </output> |

| TYPE | ORGNL. LAW EXAMPLE | RVSD. LAW EXAMPLE | ORGNL. FACT EXAMPLE | RVSD. FACT EXAMPLE | DESCRIPTION |
|---|---|---|---|---|---|
| T1.1 | *Article 271, Paragraph 2*: *Personnel of state-owned companies, enterprises, or other state-owned entities engaged in public duties, who commit acts specified in the preceding paragraph, shall be sentenced ...* | *Personnel of **private enterprises**, state-owned companies, enterprises, or other state-owned entities engaged in public duties, who commit acts specified in the preceding paragraph, shall be sentenced ...* | ... the defendant, Chen, while serving as person in charge of *company A, which is a state-owned company*, took advantage of his position to illegally appropriate client payments of a total of RMB 1.3 million... | ... the defendant, Chen, while serving as **a public duty personnel in a private enterprise B**, took advantage of his position to illegally appropriate client payments of a total of RMB 1.3 million... | The model **passes** if it predicts **Article 271, Paragraph 2** as the relevant article under the revised law. |
| T1.2 | *Article 271, Paragraph 2* | *Personnel of **any enterprise** engaged in public duties, who commit the acts specified in the preceding paragraph, shall be sentenced ...* | ... the defendant Li was employed by *the state-owned company A*, as a property assistant ... Li exploited her position and illegally embezzle parking fees totaling RMB 114,016 ... | ... the defendant Li was employed by **a private enterprise B**, as a property assistant ... Li exploited her position and illegally embezzle parking fees totaling RMB 114,016 ... | The model **passes** if it predicts **Article 271, Paragraph 2** as the relevant article under the revised law. |
| T1.3 | *Article 271, Paragraph 2* | *Personnel **in charge** of **any company or enterprise** engaged in public duties, who commit the acts specified in the preceding paragraph, shall be sentenced ...* | ... the defendant Wang took advantage of his position as a sales representative at *state-owned company A* to illegally apply for vouchers through the company's system and privately sell them... | ... the defendant Wang took advantage of his position as **sales manager at private enterprise B** to illegally apply for vouchers through the company's system and privately sell them... | The model **passes** if it predicts **Article 271, Paragraph 2** as the relevant article under the revised law. |
| T1.4 | *Article 271, Paragraph 2* | *Personnel **in charge** of **any company or enterprise** engaged in public duties, who commit the acts specified in the preceding paragraph, shall be sentenced ...* | ... the defendant Wang took advantage of his position as property rights processing clerk at *state-owned company A* to repeatedly misappropriate funds from clients, totaling RMB 646,990. | ... the defendant Wang took advantage of his position as property rights processing clerk at **private enterprise B** to repeatedly misappropriate funds from clients, totaling RMB 646,990. | The model **passes** if it does **not** predict **Article 271, Paragraph 2** as the relevant article under the revised law. |
| T1.5 | *Article 271, Paragraph 2* | *Personnel of **state-owned companies** engaged in public duties, who commit the acts specified in the preceding paragraph, shall be sentenced ...* | ... the defendant Bai took advantage of his position as head of supply station at *state-owned company A* to misappropriate RMB 908,869 ... | ... the defendant Bai took advantage of his position as head of supply station at **state-owned enterprise B** to misappropriate RMB 908,869 ... | The model **passes** if it does **not** predict **Article 271, Paragraph 2** as the relevant article under the revised law. |
| T1.6 | *Article 271, Paragraph 2* | *Personnel **in charge** of **state-owned companies** engaged in public duties, who commit the acts specified in the preceding paragraph, shall be sentenced ...* | ... the defendant Zhang, while serving as **sales administrator** at *state-owned company A* took advantage of his position to misappropriate over RMB 200,000 ... | ... the defendant Zhang, while serving as a **book sales clerk** at **state-owned company** took advantage of his position to misappropriate over RMB 200,000 ... | The model **passes** if it does **not** predict **Article 271, Paragraph 2** as the relevant article under the revised law. |
| T2.1 | *Article 354, Paragraph 1*: Those who *harbors others to inhale or inject drugs* shall be sentenced ... | Those who **harbors others to inhale, inject or trade drugs** shall be sentenced ... | ... the defendant Yu ... together with Zhang and Chen drove back to the LinLi bar owned by Yu and consumed methamphetamine together... | ... the defendant Yu ... drove Zhang and Chen back to the LinLi bar, which is Yu's property, and traded methamphetamine together ... | The model **passes** if it predicts **Article 354, Paragraph 1** as the relevant article under the revised law. |
| T2.2 | *Article 354, Paragraph 1* | Those who **harbors others to commit drug related crimes** shall be sentenced ... | ... the defendant Antonio ***provided a venue for the joint consumption of cocaine*** on four occasions at his residence to Abhimanyu ... | ... the defendant Antonio p**rovided a venue for the transactions of cocaine** on four occasions at his residence to Abhimanyu ... | The model **passes** if it predicts **Article 354, Paragraph 1** as the relevant article under the revised law. |
| T2.3 | *Article 354, Paragraph 1* | Those who **harbors others to commit drug crimes other than injecting** shall be sentenced ... | ... the defendant Huang harbored Wu and Tan on three occasions at his residence, *providing a venue for them to consume methamphetamine* ... | ... the defendant Huang harbored Wu and Tan on three occasions at his residence, **providing a venue for them to produce methamphetamine** ... | The model **passes** if it predicts **Article 354, Paragraph 1** as the relevant article under the revised law. |
| T2.4 | *Article 354, Paragraph 1* | Those who **harbors others to commit drug crimes other than injecting** shall be sentenced ... | ... the defendant Xiao repeatedly harbored his relatives Wang, Lin and Wang at his residence, *facilitating their drug trafficking activities*... | ... the defendant Xiao repeatedly harbored his relatives Wang, Lin and Wang at his residence, **facilitating their consumption of heroin through injection** ... | The model **passes** if it does **not** predict **Article 354, Paragraph 1** as the relevant article under the revised law. |
| T2.5 | *Article 354, Paragraph 1* | Those who **harbors others to inhale drugs** shall be sentenced ... | ... the defendant Wu ... *provided a venue* for his coworkers Tian and Yang where they *smoked marijuana on multiple occasions* ... | ... the defendant Wu ... **provided a venue** for his coworkers Tian and Yang where they **injected heroin on multiple occasions** ... | The model **passes** if it does **not** predict **Article 354, Paragraph 1** as the relevant article under the revised law. |

Continue to the next page

| TYPE | ORGNL. LAW EXAMPLE | RVSD. LAW EXAMPLE | ORGNL. FACT EXAMPLE | RVSD. FACT EXAMPLE | DESCRIPTION |
|---|---|---|---|---|---|
| T2.6 | *Article 354, Paragraph 1* | Those who **detains others to inhale and inject drugs** shall be sentenced ... | ... the defendant Wang consumed methamphetamine together with Li at Wang's apartment ... | ... the defendant Wang voluntarily provided a venue for Wang, with whom they consumed methamphetamine in Wang's apartment ... | The model **passes** if it does **not** predict *Article 354, Paragraph 1* as the relevant article under the revised law. |
| T3.1 | *Article 348, Paragraph 1*: Those who illegally possess more than one kilo of *opium*, more than 50g of *heroin* or *methamphetamine*, or a large quantity of *other drugs* shall be sentenced ... | Those who illegally possess more than one kilo of **opium or laundry powder**, more than 50g of *heroin* or *methamphetamine*, or a large quantity of *other drugs* shall be sentenced ... | ... the defendant Ren was found in possession of a large quantity of drugs ... during the arrest, a bag of heroin was found ... and was determined to be 145.3g ... | ... the defendant Ren was found in possession of a large quantity of laundry powder ... during the arrest, a bag of laundry powder was found ... and was determined to be 2 kilos ... | The model **passes** if it predicts *Article 348, Paragraph 1* as the relevant article under the revised law. |
| T3.2 | *Article 348, Paragraph 1* | Those who illegally possess a large amount of **addictive substances** shall be sentenced ... | ... the defendant Hu was found in possession of 2.3 kilos of opium at his house arrest ... | ... the defendant Hu was found in possession of 2.3 kilos of alcohol at his house arrest ... | The model **passes** if it predicts *Article 348, Paragraph 1* as the relevant article under the revised law. |
| T3.3 | *Article 348, Paragraph 1* | Those who illegally possess a large amount of **poppy-based products or injectable drugs** shall be sentenced ... | ... the defendant Wei ... was arrested ... upon arrest, he was found with possession of 2 kilos of opium ... | ... the defendant Wei ... was arrested ... upon arrest, he was found with possession of 2 kilos of poppy seed oil ... | The model **passes** if it predicts *Article 348, Paragraph 1* as the relevant article under the revised law. |
| T3.4 | *Article 348, Paragraph 1* | Those who illegally possess a large amount of **poppy-based products or injectable drugs** shall be sentenced ... | ... the defendant Liu was found with a total of 1.7 kilos of opium ... | ... the defendant Liu was found with a total of 1.7 kilos of marijuana ... | The model **passes** if it does **not** predict *Article 348, Paragraph 1* as the relevant article under the revised law. |
| T3.5 | *Article 348, Paragraph 1* | Those who illegally possess more than **50g of methamphetamine** shall be sentenced ... | ... the defendant Chen was found in possession of 60g *methamphetamine* at his house arrest ... | ... the defendant Chen was found in possession of 60g **heroin** at his house arrest ... | The model **passes** if it does **not** predict *Article 348, Paragraph 1* as the relevant article under the revised law. |
| T3.6 | *Article 348, Paragraph 1* | Those who illegally possess more than one kilo of *opium*, more than 50g of *heroin* or *methamphetamine*, or a large quantity of **other synthetic drugs** shall be sentenced ... | ... the defendant Liang was driving with 255 *methamphetamine tablets* when he was intercepted by police at the checkpoint ... | ... the defendant Liang was driving with 2 kilos of **marijuana** when he was intercepted by police at the checkpoint ... | The model **passes** if it does **not** predict *Article 348, Paragraph 1* as the relevant article under the revised law. |
| T4.1 | *Article 213, Paragraph 1*: Those who use a trademark identical to a registered trademark on the same goods or services *without the permission of the trademark owner* ... shall be sentenced ... | Those who use a trademark identical to a registered trademark on the same goods or services **without the permission of the trademark owner or the owner's parents** ... shall be sentenced ... | ... the defendant Wang, in collaboration with Chen rented a factory to produce bulk laundry detergent under the registered trademarks A and B *without the owner's permission* ... | ... the defendant Wang, in collaboration with Chen rented a factory to produce bulk laundry detergent under the registered trademarks A and B **with the owner's mother's permission** ... | The model **passes** if it does **not** predict *Article 213, Paragraph 1* as the relevant article under the revised law. |
| T4.2 | *Article 213, Paragraph 1* | Those who use a trademark identical to a registered trademark on the same goods or services **without the permission of members of the trademark owner's family of origin** ... shall be sentenced ... | ... the defendant Wang, in collaboration with Chen rented a factory to produce bulk laundry detergent under the registered trademarks A and B *without the owner's permission* ... | ... the defendant Wang, in collaboration with Chen rented a factory to produce bulk laundry detergent under the registered trademarks A and B **with the owner's mother's permission** ... | The model **passes** if it does **not** predict *Article 213, Paragraph 1* as the relevant article under the revised law. |
| T4.3 | *Article 213, Paragraph 1* | Those who use a trademark identical to a registered trademark on the same goods or services **without the permission of the registered trademark owner's direct relatives within three generations** ... shall be sentenced ... | ... the defendant Wang, in collaboration with Chen rented a factory to produce bulk laundry detergent under the registered trademarks A and B *without the owner's permission* ... | ... the defendant Wang, in collaboration with Chen rented a factory to produce bulk laundry detergent under the registered trademarks A and B **with the owner's aunt's permission** ... | The model **passes** if it does **not** predict *Article 213, Paragraph 1* as the relevant article under the revised law. |
| T4.4 | *Article 213, Paragraph 1* | Those who use a trademark identical to a registered trademark on the same goods or services **without the permission of the registered trademark owner's direct relatives within three generations** ... shall be sentenced ... | ... the defendant Wang, in collaboration with Chen rented a factory to produce bulk laundry detergent under the registered trademarks A and B *without the owner's permission* ... | ... the defendant Wang, in collaboration with Chen rented a factory to produce bulk laundry detergent under the registered trademarks A and B **with the owner's permission** ... | The model **passes** if it predicts *Article 213, Paragraph 1* as the relevant article under the revised law. |

Continue to the next page

| TYPE | ORGNL. LAW EXAMPLE | RVSD. LAW EXAMPLE | ORGNL. FACT EXAMPLE | RVSD. FACT EXAMPLE | DESCRIPTION |
|---|---|---|---|---|---|
| T4.5 | *Article 213, Paragraph 1* | Those who use a trademark identical to a registered trademark on the same goods or services **without the written permission of the registered trademark owner** ... shall be sentenced ... | ... the defendant Wang, in collaboration with Chen rented a factory to produce bulk laundry detergent under the registered trademarks A and B *without the owner's permission* ... | ... the defendant Wang, in collaboration with Chen rented a factory to produce bulk laundry detergent under the registered trademarks A and B **with the owner's oral permission** ... | The model **passes** if it predicts *Article 213, Paragraph 1* as the relevant article under the revised law. |
| T4.6 | *Article 303, Paragraph 2*: Those who opens a casino shall be sentenced ... | Those who opens a casino **without the approval from People's Bank of China** shall be sentenced ... | ... the defendant opens a casino ... | ... the defendant opens a casino with the approval from People's bank of China ... | The model **passes** if it does **not** predict *Article 213, Paragraph 1* as the relevant article under the revised law. |
| T4.7 | *Article 213, Paragraph 1* | Those who use a trademark identical to a registered trademark on the same goods or services ... shall be sentenced ... | ... the defendant Wang, in collaboration with Chen rented a factory to produce bulk laundry detergent under the registered trademarks A and B *without the owner's permission* ... | ... the defendant Wang, in collaboration with Chen rented a factory to produce bulk laundry detergent under the registered trademarks A and B **with the owner's permission** ... | The model **passes** if it predicts *Article 213, Paragraph 1* as the relevant article under the revised law. |
| T5.1 | *Article 274, Paragraph 1*: Those who extort public or private property ... shall be sentenced ... | Those who extort public or private property **for the purpose of revenge** ... shall be sentenced .... | ... the victim Li built a house, contracted to Xia, and was forced to pay the defendnat Zhang 9,000 yuan after Zhang obstructed the construction ... | ... the victim Li built a house, and was forced to pay Zhang 9,000 yuan after Zhang obstructed the construction *for the purpose of causing chaos* ... | The model **passes** if it does **not** predict *Article 274, Paragraph 1* as the relevant article under the revised law. |
| T5.2 | *Article 224, Paragraph 1*: Whoever, *with the intent of illegal possession*, defrauds the other party of property during the signing or performance of a contract under any of the following circumstances, shall be sentenced ... | Whoever, defrauds the other party of property during the signing or performance of a contract under any of the following circumstances, shall be sentenced ... | ... the defendant Qi, *with the intent of illegal possession*, engaged in a fraudulent scheme ... through this method, he fraudulently obtained steel worth RMB 721,375.41 ... | ... the defendant Qi, **with the intent of revenge**, engaged in a fraudulent scheme ... through this method, he fraudulently obtained steel worth RMB 721,375.41 ... | The model **passes** if it predicts *Article 274, Paragraph 1* as the relevant article under the revised law. |
| T6.1 | *Article 232, Paragraph 1*: Whoever intentionally commits homicide shall be sentenced to *death, life imprisonment, or at least 10 years' imprisonment*. For relatively minor cases, the sentence shall be *3 to 10 years' imprisonment*. | Whoever intentionally commits homicide shall be sentenced to *death, life imprisonment, or at least 10 years' imprisonment*. For relatively minor cases, the sentence shall be **more than 10 years' imprisonment**. | - | - | If, after the law revision, the model predicts at least 10 years of imprisonment for cases originally sentenced to 3–10 years, it is considered a **pass**. |
| T6.2 | *Article 232, Paragraph 1* | Whoever intentionally commits homicide shall be sentenced to *death, life imprisonment, or at least 10 years' imprisonment*. For relatively minor cases, the sentence shall be **less than 3 years' imprisonment**. | - | - | If, after the law revision, the model predicts no more than 3 years of imprisonment for cases originally sentenced to 3–10 years, it is considered a **pass**. |
| T6.3 | *Article 232, Paragraph 1* | Whoever intentionally commits homicide shall be sentenced to *death, life imprisonment, or at least 10 years' imprisonment*. For relatively minor cases, the sentence shall be **death or life imprisonment**. | - | - | If, after the law revision, the model predicts death or life imprisonment for cases originally sentenced to 3–10 years, it is considered a **pass**. |
| T6.4 | *Article 232, Paragraph 1* | Whoever intentionally commits homicide shall be sentenced to **at least 10 years' imprisonment**. For relatively minor cases, the sentence shall be *3 to 10 years' imprisonment*. | - | - | If, after the law revision, the model predicts more than 10 years of imprisonment for cases originally sentenced to death or life imprisonment, it is considered a **pass**. |

Table 13: Detailed examples and descriptions of every revision type.

