# OpenReview forum: "LawShift: Benchmarking Legal Judgment Prediction Under Statute Shifts"
_NeurIPS.cc/2025/Datasets_and_Benchmarks_Track — NeurIPS 2025 Datasets and Benchmarks Track poster_

### Official Review · Reviewer_kpWV · 2025-06-28

**Rating:** 6
**Confidence:** 4

**Summary:**

The paper proposes LawShift, a dataset designed to evaluate the reliability of legal judgment prediction (LJP) when faced with legal updates. Using this dataset, the authors tested and identified certain limitations of SOTA LJP models.

**Additional Feedback:**

I have doubts about the meaning of the "normalized relative pass rate shift" metric. Literally, this metric should reflect the extent of change in the LJP model's pass rate when faced with legal revisions, which suggests it should involve the absolute difference between the pass rates before and after the revisions. Intuitively, a smaller change in pass rate would indicate greater adaptability of the LJP model. However, in the experimental analysis, the authors seem to interpret a lower value of this metric as indicative of poorer model performance. I hope the authors can clarify this concern.

**Dataset Code Accessibility:**

Yes

**Dataset Code Comments:**

The dataset has been released to HuggingFace, and the code is provided via an Anonymous GitHub link.

**Ethical Comments:**

Despite a study on laws, no ethical concerns are found in this paper.

**Ethical Considerations:**

No, there are no or only very minor ethics concerns

**Final Justification:**

Despite a few issues (assumptions and evaluations, etc.) raised in the rebuttal and discussion, they are finally clarified and my concerns are addressed. As such, I would keep my original rating.

**Limitations Weaknesses:**

W1: The authors selected criminal judgments issued between November 4, 2017, and February 28, 2021, to ensure consistency in the legal versions across all cases in the dataset. However, based on the principle of non-retroactivity of law, I believe cases should be selected according to whether the criminal acts occurred during the same period under a consistent legal version.

W2: Only one LLM is evaluated in the experiments. Including additional general-purpose and legal-domain LLMs would be beneficial for better understanding the strengths and limitations of SOTA LJP models.

W3: The design and meaning of the "normalized relate pass rate" metric should be further clarified.

**Strengths Contributions:**

S1: This paper is easy to follow and clearly describes the details of the benchmark design.

S2: The paper targets a research problem that is important in legal contexts but has been largely underexplored.

S3: LawShift is the first dataset designed to evaluate the reliability of LJP in the context of law updates.

S4: The benchmark design of LawShift is highly detailed and deliberated. The authors employed three legal experts to develop a law article template that structures the components of legal articles and, based on both the template and historical editions of Chinese Criminal Code, derived a comprehensive set of law revision types. This enables users to systematically evaluate the reasoning reliability of LJP models when confronted with different types of legal updates.

S5: The experimental results are highly insightful, highlighting the urgent need for developing adaptable LJP approaches in dynamic legal contexts.

---

> ### Author Rebuttal · Authors · 2025-07-31
>
> We appreciate the reviewer's thoughtful and valuable comments. Below, we respond to each comment one by one.
>
> ### **Clarification on Legal Version Consistency**
>
> Thank you for the valuable observation. In Appendix B (lines 554–559, lines 565-567), we provide a detailed explanation for selecting cases dated between November 4, 2017, and February 28, 2021, to ensure that all criminal acts fall within the same period under a consistent legal version. We will clarify this point in the final version of the paper.
>
>
> ### **Model Evaluation and Future Extensions**
>
> Thank you for the helpful suggestion. We agree that evaluating additional LLMs would provide a more comprehensive understanding of SOTA models' strengths and limitations. In the revised version, we will extend our evaluation to include both general-purpose LLMs (e.g., GPT-4, Claude, or Gemini, subject to API availability) and domain-specific legal LLMs (e.g., Legal-BERT-based variants, ChatLaw,). We will report performance across these models using the same testbed, and discuss how model specialization influences reasoning and judgment capabilities.
>
> ### **Clarifying the "Normalized Relative Pass Rate Shift" Metric**
>
> Thank you for pointing out the potential confusion. The Normalized Relative Pass Rate Shift (NR-PRS) is designed not to measure the raw drop in accuracy for a single revision type, but to compare how strongly the same component (subject, action, etc.) disrupts different models after we normalize for each model’s own baseline level.
>
> - Definition
>
> For a given component $c$ with revision sub-types, we compute:
>
> $ \text{NR-PRS}\_{c} = \sum\_{i=1}^{n\_{c}}\log\Bigl(2 +\frac{r^{\text{rev}}\_{i} -r^{\text{orig}}\_{i}}{r^{\text{orig}}\_{i}}\Bigr) $
>
> where $ r^{orig}_i $ and $ r^{rev}_i $ are the model’s pass rates before and after the revision.
> The ratio is therefore a relative change; the + 2 inside the log is a stability term that prevents negative or zero arguments when a model performs poorly at baseline.
>
> - Why higher NR-PRS means greater robustness
>
> If a model suffers almost no degradation on any sub-type, each fraction $ (r^{rev} - r^{orig})/r^{orig} $ is close to zero. The log term approaches log 2; summed over all sub-types, the NR-PRS is high.
> When the pass rate drops sharply, the fraction is negative, the logarithm shrinks, and the sum falls. Thus a larger NR-PRS indicates that the model exhibits proper level of adaptability when faced with law revisions. Conversely, a smaller value (directly derived from significantly low $ r^{rev}$ ) flags severe post-revision degradation.
>
> - Why we prefer a relative, log-stabilized measure
>
> Absolute percentage drops do not account for the model’s starting point: Lawformer, for instance, may begin at 92 % on T1.1 but 100.0% on T2.1. A same 10% pass rate drop therefore reflects very different levels of disruption. By dividing each drop by the corresponding original pass rate and then taking the logarithm, the metric removes this baseline bias and lets us compare revision impact across components without the result being skewed by how accurate the model was to begin with.
> Thank you for the suggestion. We will clarify the definition and usage of NR-PRS in the final version.
>
>
> We sincerely thank the reviewer for highlighting these important considerations.

---

> > ### Comment · Reviewer_kpWV · 2025-08-04
> > **Re: Author Rebuttal**
> >
> > Thanks for the authors' effort.
> >
> > **Normalized Relative Pass Rate Shift:** The authors stated that "when the pass rate drops sharply, the fraction is negative," which assumes that $r_i^{rev}$ will not exceed $r_i^{orig}$. I remain skeptical of this assumption; although it could hold in this paper's experiments, such a design may mislead readers. A more appropriate approach would be to compute the absolute difference between $r_i^{rev}$ and $r_i^{orig}$.
> >
> > Furthermore, the authors argued that the use of a logarithmic form is intended to mitigate bias caused by the denominator $r_i^{orig}$. If the concern is merely to prevent extreme values due to small $r_i^{orig}$, a more conventional and justifiable alternative would be $2*|r_i^{rev} - r_i^{orig}|/(r_i^{rev} + r_i^{orig})$. The use of the log2p formulation lacks clear justification.
> >
> > **Model Evaluation:** Since the authors did not choose to include additional LLMs, I cannot be fully convinced by the experimental conclusions.

---

> > > ### Author Response · Authors · 2025-08-04
> > > **Response to Model Evaluation**
> > >
> > > We apologize for initially interpreting the request for additional LLM experiments as a suggestion rather than a requirement. We are actively conducting the experiments. We truly appreciate your understanding and kindly ask for approximately 1-2 days to finalize the results and integrate them into our response. We remain committed to thoroughly addressing your concerns and ensuring the response meets your expectations.

---

> > > > ### Comment · Reviewer_kpWV · 2025-08-07
> > > > **Re: Author Response**
> > > >
> > > > I sincerely appreciate the authors for adopting the metric I suggested, and I look forward to seeing the new experimental results on additional LLMs.
> > > >
> > > > However, I still have a concern: why should models with a higher pass rate after revision receive a higher reward? Since the authors state that their goal is robustness assessment, should robustness not  suggest consistent performance after law revisions? In legal practice, professionals are likely to prefer models whose predictive capabilities can be clearly anticipated, rather than those whose accuracy fluctuates significantly after law revisions. Therefore, even if accuracy improves after law revisions, I am wondering if this should be rewarded.

---

> > > > > ### Author Response · Authors · 2025-08-07
> > > > > **Response to more LLM experiments (3/3)**
> > > > >
> > > > > We also report each model’s normalized relative pass rate shift across the six components in the below table. The score is calculated as $\text{NR-PRS}\_c = \sum^{n_c}_{i=1} 2* (r^{rev_i} - r^{orig_i})/(r^{rev_i} + r^{orig_i})$.
> > > > >
> > > > > |Component |   Llama3-Chinese-8B | ChatGLM3-6B |Llama3-8B-Chinese-LoRA-Law   |   Law-model-7B  |  Qwen2.5-7B|
> > > > > |-----|-----|-----|-----|-----|-----|
> > > > > |subject  |  -0.38454 |-0.00651| -0.57625 | 0.61313| -0.21058|
> > > > > |action   |  -1.01195 | 0.44797| -1.01965| -0.39020 |-0.97567|
> > > > > |object   |   0.19604|  0.53013| -0.81756| -0.19351|  0.84329|
> > > > > |objCon   |   0.00060 | 0.40753|  0.03406| -0.16586 | 0.62442|
> > > > > |sbCon   |   0.26179 | 0.76023|  0.70060 | 0.28472 | 0.68693|
> > > > > |term    |   -0.00921|  0.34706 |-0.11936  |0.55424| -0.26669|
> > > > >
> > > > > Across all components, action revisions remain the most challenging for each LLM. Llama3-Chinese-Instruct and Llama3-Chinese-LoRA-Law suffer the largest negative shifts on action revisions. Qwen also shows a marked drop in this category compared to other categories. ChatGLM3-6B and Qwen see notable positive scores on object revisions, while Llama3-Chinese-LoRA-Law and Law-model-7B’s scores are negative, suggesting that models pretrained on legal texts may be less flexible in the face of revisions in the object component. Law-model-7B shows the highest adaptability to revisions involving the subject and term components.
> > > > >
> > > > > To better explain LLMs’ poor performance on action revisions, we present a representative failure case involving T2.6, where models are expected to distinguish between coercive actions (“detaining others to inject or inhale drugs”, which should lead to a guilty verdict) and non-coercive actions (“harboring others who voluntarily inject or inhale drugs”, which should not). However, the models tend to conflate the two actions, leading to misclassifications. This failure points to a gap in legal semantic understanding and reasoning. First, facts of actions are often spread across multiple sentences, requiring the model to integrate dispersed information. Second, while LLMs may grasp general semantics, they often lack awareness of legal definitions. For instance, understanding that “detaining” implies coercion under the law. These misinterpretations indicate that current models rely heavily on surface-level semantics, lacking a grasp of the legal meaning conveyed in natural language. Enhancing performance on law revisions requires deeper understanding of legal semantics and reasoning, especially when judgments depend on subtle distinctions in actions.
> > > > >
> > > > > [1] Jinze Bai, et al. "Qwen technical report." arXiv preprint arXiv:2309.16609 (2023).
> > > > >
> > > > > [2] Yiming Cui, Ziqing Yang, and Xin Yao. "Efficient and effective text encoding for Chinese llama and alpaca (2023)." arXiv preprint arXiv:2304.08177.
> > > > >
> > > > > [3] Aohan Zeng, et al. "Chatglm: A family of large language models from glm-130b to glm-4 all tools." arXiv preprint arXiv:2406.12793 (2024).
> > > > >
> > > > > [4] https://ollama.com/mrhua/llama3-8b-chinese-lora-law_f16_q4_0/blobs/b14d71cdfffa
> > > > >
> > > > > [5] https://ollama.com/initium/law_model
> > > > >
> > > > > [6] Fei, Zhiwei, et al. "Lawbench: Benchmarking legal knowledge of large language models." arXiv preprint arXiv:2309.16289 (2023).
> > > > >
> > > > > Thank you again for the thoughtful comment.

---

> > > > > ### Author Response · Authors · 2025-08-07
> > > > > **Response to robustness**
> > > > >
> > > > > We appreciate the reviewer’s comments regarding model robustness. In our work, we define robustness as the model’s ability to make correct predictions even when statutes change. That is, the model can correctly adapt its judgment to reflect the revised statutes. It is not robustness that produces identical predictions before and after revisions. In the legal domain, such invariance is not necessarily desirable. Revisions to statutes are often made precisely to change legal outcomes, and a robust model should be able to adjust its predictions accordingly. If a model maintains the same prediction after a revision, it would mean the model is insensitive to law. What legal professionals truly require is not invariance in predictions, but consistency with the applicable law. Therefore, our evaluation focuses on whether the model can make correct predictions under the revised statutes, rather than maintaining identical predictions regardless of statutory change.
> > > > >
> > > > > But the reviewer’s concern about LJP tools’ predictability and stability inspires us a lot. It highlights an important trade-off in the design of legal AI systems: ensuring that models remain responsive to changes in law, while also maintaining transparent and interpretable behavior that legal professionals can trust. In future work, we plan to further explore how to enhance the transparency of LJP models, enabling users to understand their reasoning processes and gain insights into their underlying decision-making mechanisms.

---

> > > > > > ### Comment · Reviewer_kpWV · 2025-08-08
> > > > > >
> > > > > > Thanks to the authors for providing such extensive experimental results, which addressed my major concern.
> > > > > >
> > > > > > Regarding model robustness, the authors may have misunderstood my comment: What I mean is that the model's predictive capability does not change significantly after law revisions, rather than that its prediction results remain unchanged. They are different things. Nonetheless, despite different views on the definition of model robustness, this is not a big issue as long as the authors can clearly justify their metric in the paper, and ideally report experimental results under both definitions (i.e., the authors’ and mine). Therefore, I will maintain my original score.

---

> > ### Author Response · Authors · 2025-08-04
> > **Response to Normalized Relative Pass Rate Shift (1/2)**
> >
> > Thank you for the feedback and the alternative metric suggestion. We address the three concerns in turn.
> >
> > 1. Assumption about $ r^{rev} $ vs. $ r^{orig} $
> >
> > Our formulation $ \text{NR-PRS}\_{c} = \sum\_{i=1}^{n\_{c}}\log\Bigl(2 +\frac{r^{\text{rev}}\_{i} -r^{\text{orig}}\_{i}}{r^{\text{orig}}\_{i}}\Bigr) $ makes NO assumption that $ r^{rev} $ < $ r^{orig} $.
> > - When $ r^{rev} $ > $ r^{orig} $, the term inside the log is simply larger than 2, producing a larger (positive) score;
> > - When$ r^{rev} $ < $ r^{orig} $, it falls between 1 and 2, yielding a smaller score.
> > This monotonic behavior matches our design goal: models with higher pass rate after revision receive a higher reward, while those that do not receive a lower score.
> >
> > 2. Why an absolute-value shift is undesirable for our goal
> >
> > However, the absolute value makes the metric **direction-blind**: an improvement and an equal degradation receive the same score, and in more extreme circumstances, significant degradation receives a higher score than mild improvements. An example is shown in the following table, this is the pass rate of our Qwen model on T1.1 and T1.4 (detailed statistics in Table 10 of our original manuscript), respectively:
> > |Type|Trend|$r^{orig}$ (%)|$r^{rev}$ (%)|Absolute-value version|Ours|
> > |----|-------|-----|-----|----|-----|
> > |T1.1|Improvement|59.00|78.54|0.2841|$log\_2 (2.3312) $ ≈ 1.2211|
> > |T1.4|Degradation|59.00|29.12|0.6782|$log\_2 (1.4936) $ ≈ 0.5788|
> >
> >
> > Under a metric that should reward improvement, T1.1 ought to score higher than T1.4. Our formulation does so, whereas the absolute-value version assigns the *degraded* result in a larger score, contradicting intuitive expectations.
> >
> > Our objective is robustness assessment, where a model that maintains or improves its pass rate should be rewarded, while a drop should be penalised. Direction therefore matters.
> >
> > 3. Rationale for the log(2+) transformation
> >
> > We chose the log-stabilized score because raw pass-rate ratios suffer from two practical problems that distort comparisons:
> > |Requirement|Issue with raw ratio|How log(2+) fixes it|
> > |--------|---------|---------|
> > |Reduce influence of extreme spikes or drops|When r^orig is very small, even a modest absolute change can explode to a huge ratio, letting a single outlier dominate the sum|**The logarithm compresses large values while preserving order, so no single item overwhelms the aggregated sum**|
> > |Keep a finite, well defined domain|The ratio $(r^{rev}-r^{orig})/r^{orig}$ explodes as $ r^{orig} \rightarrow $  0; can be -1 when $ r^{rev} $=0| Adding the +2 term shifts the argument to [1, +∞), ensuring legal inputs to the log.|
> >
> > The log term reins in cases where an extremely small $r^{orig}$ would otherwise yield an outsized “improvement” purely because the denominator is tiny, leveraging the property of the logarithm that its growth rate diminishes as its argument increases.
> >
> > The reviewer proposed replacing the plain ratio with a denominator that automatically scales the change by the combined magnitude of the two runs. This form naturally suppresses outliers, needs no additive constant, and remains strictly monotonic over the full range of values. We agree that it is an elegant improvement of our earlier log-based transformation and will adopt it in the revised paper.

---

> > ### Author Response · Authors · 2025-08-04
> > **Response to Normalized Relative Pass Rate Shift (2/2)**
> >
> > Below, we present the statistics under both metrics. Due to the constraints of the rebuttal format, we include the original average scores for each component. The results under the new metric exhibit a similar overall trend.
> >
> > - $\text{NR-PRS}\_c = \sum^{n_c}_{i=1} 2* (r^{rev_i} - r^{orig_i})/(r^{rev_i} + r^{orig_i})$
> >
> > |Component|Top|Ladan|Neur|Law|Qwen|
> > |--------|---------|--------|--------|--------|--------|
> > |subject|-0.83191|-0.74365|-0.81010|-0.80813|-0.21058|
> > |action|-0.96132|-0.81611|-0.84222|-1.21755|-0.97567|
> > |object|-0.99067|-0.76340|-0.81133|-0.83830 |0.84329|
> > |objCon|-0.92795|-0.93219|-0.93254|-0.91220 |0.62442|
> > |sbCon|-0.65465|-0.71140|-0.79851|-0.73335|0.68693|
> > |term| -0.86342| -0.97731 |-1.10237 |-1.04387| -0.26669|
> >
> > - $ \text{NR-PRS}\_{c} = \sum\_{i=1}^{n\_{c}}\log\Bigl(2 +\frac{r^{\text{rev}}\_{i} -r^{\text{orig}}\_{i}}{r^{\text{orig}}\_{i}}\Bigr) $
> > |Component|Top|Ladan|Neur|Law|Qwen|
> > |--------|---------|--------|--------|--------|--------|
> > |subject  |  0.54561 |  0.60203 |  0.56376 |  0.56740 |  0.87971|
> > |action  |   0.49510 |  0.54445|   0.53238 |  0.37419 |  0.49936|
> > |object  |   0.48267  | 0.58659 | 0.56062 |  0.55589 |  2.17370|
> > |objCon  |   0.52145 |  0.52611 |  0.54670 |  0.53239  | 1.98429|
> > |sbCon   |   0.63962 |  0.61241|   0.56439|   0.60322 |  1.83773|
> > |term   |    0.73239 |  0.46478 |  0.52525 |  0.43674  | 0.87081|
> >
> >
> > The two metrics exhibit nearly identical trends. Notably, the TopJudge score for the term component (line 302) is less inflated under the new metric. This suggests that the new metric is more effective at mitigating score inflation caused by a small denominator.
> >
> > Comparing the two tables, it is evident that:
> >
> > (1) Sign consistency holds. The sign of every raw average score is unchanged between the two calculations: scores that are negative under the new metric are all the scores that are less than 1 the old metric.
> >
> > (2) Component ordering is identical. In both metrics, for example, object and objCon give Qwen the largest positive shift, while all four traditional models remain negative; action is the steepest drop for every model; and term sits low across the board.
> >
> > (3) Model ranking is preserved. Whenever a model is the best (or worst) under the log metric, it is also best (or worst) under the new metric. Top stays lowest on Object and objCon, and Qwen consistently tops the positive side.
> >
> > These one-to-one correspondences confirm that the alternative normalisation changes only the numeric scale, not the relative tendencies, so the visual “shape” of the line chart (Fig. 4) remains virtually unchanged, which means our conclusions still hold.
> >
> > We again thank the reviewer for the suggestions and feedback! If there are still any concerns, we are eager to clarify them.

---

> > ### Author Response · Authors · 2025-08-07
> > **Response to more LLM experiments (1/3)**
> >
> > We thank the reviewer’s suggestion again for incorporating more LLMs experiments. In our original experiments we evaluated Qwen [1] as it achieves SOTA results on legal tasks [6].  In the updated experiments, we included four additional LLMs: two general-domain and two legal-domain models.
> >
> > (1) Llama-3-Chinese-8B-Instruct [2]: A general-domain variant of the Llama 3 family, instruction-tuned on a large multilingual corpus with a strong focus on Chinese.
> >
> > (2) ChatGLM3-6B [3]: A general-domain open-source conversational model of the ChatGLM family. Trained on massive bilingual chat and knowledge data, it offers strong dialogue coherence, particular in Chinese.
> >
> > (3) Llama3-8B-Chinese-LoRA-Law [4]: A legal-domain LLM. Built by applying LoRA to the base Llama3-8B-Chinese model, fine-tuned on a specialized legal corpus of statutes, case law, and judicial opinions.
> >
> > (4) Law-model-7B [5]: A legal-domain LLM. The model is fine-tuned on Mistral 7B and specializes in the legal domain.
> >
> > We employ a two-phase strategy to incorporate revisions into the LLMs for generating predictions. First, each model is “informed” with the full set of applicable statutes, either the original or the revised versions. In this way, the model would internalize the statutes. Next, for each case we present only the fact description and instruct the model to output a fixed three-line response containing the charge label, article label, and penalty term. All inferences are executed locally via the Ollama framework with deterministic decoding (temperature=0). The prompts used are:
> > |Initial Prompt (Translated from Chinese)|
> > |-----|
> > |You are a legal judgment prediction tool. Based on the following case facts, you will refer to the relevant legal statutes to produce a judgment, including the law article label, the charge label, and the sentence term. Below are the statutes you should study, formatted as “Article Label: [Charge] Text”, … [list of statutes omitted here] …. Please review the statutes above. I will then provide the case facts for you to judge. At this moment, you do not need to return any output.|
> >
> > |Judgment Prompt (Translated from Chinese)|
> > |------|
> > |You are a legal judgment prediction tool. Based on the following case facts, please predict the charge, relevant statute, and penalty terms. The content after [fact] is the case’s fact description. [fact] … [real fact ommited here] …Please strictly follow this format when you output your answers and do not output anything else. Statute label: a-b-c\nCharge label: xxx\nPenalty term: n\n Please choose your statute label prediction among these: … [statute label set omitted here] … Choose your charge label prediction among these: … [charge label set omitted here] … Output the prison term in months; when life imprisonment is sentenced, output 10000; when death penalty is sentenced, output 10001.|
> >
> > We evaluate performance using pass rate, defined as the percentage of cases in which the model’s predicted statute (for all revision types other than T6.x) or the predicted penalty term (for T6.x) match the ground truth under each revision type. Details of what is considered a pass for each revision type is shown in Table 12 (page 19-21) in our paper.
> >
> > We report the pass rate of each revision in the following table, and include Qwen here for comparisons.
> >
> > |Type|Llama3-Chinese-8B-Instruct|ChatGLM3-6B|Llama3-8B-Chinese-LoRA-Law|Law-model-7B|Qwen2.5-7B|
> > |-----|-----|-----|-----|-----|-----|
> > |Ori.1|65.90|47.13|74.71|19.92|59.00|
> > |T1.1|72.41|32.95|84.29|20.69|78.54|
> > |T1.2|72.03|32.95|84.29|19.92|74.71|
> > |T1.3|74.71|32.57|82.38|21.07|56.70|
> > |T1.4|31.03|64.37|19.54|78.16|29.12|
> > |T1.5|25.67|68.20|16.86|80.08|41.38|
> > |T1.6|21.46|67.82|14.18|78.93|27.97|
> > |Ori.2|80.39|31.37|91.18|75.49|97.06|
> > |T2.1|33.33|45.10|77.45|57.84|100.0|
> > |T2.2|30.39|20.59|67.65|49.02|93.14|
> > |T2.3|30.39|36.28|72.55|62.75|81.37|
> > |T2.4|43.14|81.37|5.88|41.18|8.82|
> > |T2.5|24.51|74.51|5.88|53.92|0.0|
> > |T2.6|4.90|82.35|1.96|43.14|0.0|
> > |Ori.3|37.69|14.81|88.02|59.26|10.24|
> > |T3.1|33.55|11.33|91.94|53.59|11.11|
> > |T3.2|32.68|8.28|90.41|52.51|11.76|
> > |T3.3|32.90|11.33|90.41|52.51|11.11|
> > |T3.4|63.18|88.02|8.06|47.06|88.89|
> > |T3.5|68.19|89.32|6.97|47.06|88.45|
> > |T3.6|62.09|88.02|8.93|41.18|88.89|
> > |Ori.4|40.67|24.00|40.00|50.67|23.33|
> > |T4.1|58.00|78.00|42.67|49.33|76.67|
> > |T4.2|58.00|78.00|43.33|48.67|76.00|
> > |T4.3|56.67|80.00|42.00|46.67|76.67|
> > |T4.4|46.67|18.00|58.67|53.33|21.33|
> > |T4.5|46.00|18.67|57.33|51.33|22.67|
> > |Ori.4.6|87.04|18.58|79.22|24.35|1.96|
> > |T4.6|12.47|81.91|17.36|25.22|97.78|
> > |Ori.4.7|40.94|24.16|40.27|75.79|23.49|
> > |T4.7|44.97|16.78|53.69|26.16|24.16|
> > |Ori.5.1|36.81|17.45|36.38|37.02|15.32|
> > |T5.1|65.11|88.09|57.45|64.04|82.55|
> > |Ori.5.2|55.65|4.35|9.57|24.35|0.00|
> > |T5.2|53.91|5.22|26.96|25.21|0.00|
> > |Ori.6|0.93|7.91|20.47|3.72|29.30|
> > |T6.1|0.0|3.26|1.40|3.72|13.02|
> > |T6.2|100.0|93.49|97.67|94.88|53.95|
> > |T6.3|0.0|2.33|0.0|0.47|13.02|
> > |Ori.6.4|0.0|10.10|1.01|2.02|53.54|
> > |T6.4|100.0|96.97|98.99|97.98|47.47|

---

> > ### Author Response · Authors · 2025-08-07
> > **Response to more LLM experiments (2/3)**
> >
> > **1. LLMs-based vs. Neural-based models** (refer to Table 10, page 17 for neural-based model results). Compared to LLMs, neural-based models tend to perform worse on revision types that require changes in judgment predictions (e.g., T1.3-T1.6, T4.1-T4.3, and T6.4). Neural-based LJP models are highly sensitive to statistical co-occurrence patterns between case facts and legal judgments learned during training, rather than explicitly modeling the legal reasoning relationship between them. As a result, they tend to “stick” to judgment outcomes they’ve seen during training, even when the input has changed in a way that should affect the prediction. However, LLMs rely on generative or instruction-following strategies. This makes them better equipped to legal revisions when provided with statute amendments or factual changes. Their flexible internal representations allow them to revise legal reasoning. Additionally, LLMs are pretrained on massive corpora, and they may see more patterns between case facts and legal judgments and less influenced by label frequencies. Yet the LLMs’ advantage vanishes on types that don’t require changes in judgment predictions (e.g., T3.1-T3.3). This further suggests that neural-based models are prone to overfitting to training datasets, whereas LLMs exhibit more flexible reasoning.
> >
> > **2. General LLMs vs. Legal LLMs.** There is an evident gap between general LLMs and legal LLMs. For example, Llama3-8B-Chinese-LoRA-Law achieves near-perfect performance on revision types T3.1–T3.3 (non-changed predictions), comparable to that of neural-based models. Its general-domain variant does not exhibit comparable performance. We attribute this to its fine-tuning on extensive legal corpora, which gives it richly calibrated statutory semantics and precedent awareness but more overfits to training patterns. It causes models to prioritize training patterns over contextual legal reasoning. Conversely, legal-domain LLMs struggle to adapt when outcome changes are necessary (e.g., T3.4–T3.6 and T4.1–T4.3).

---

### Official Review · Reviewer_xzpX · 2025-06-30

**Rating:** 4
**Confidence:** 4

**Summary:**

LawShift is a new benchmark dataset specifically designed to evaluate the adaptability of Legal Judgment Prediction (LJP) models in scenarios where legal texts are dynamically revised. The dataset covers 31 types of fine-grained legal revisions, providing resources for systematically evaluating the performance of existing SOTA models under legal changes. Through systematic testing of five representative LJP models, LawShift revealed that existing models have significant limitations in responding to legal updates. The study also found that the model structure plays a key role in adaptability, and provides feasible ideas and references for future LJP research in a dynamic legal environment.

**Additional Feedback:**

See Limitations Weaknesses.

**Dataset Code Accessibility:**

Yes

**Dataset Code Comments:**

See Strengths Contributions.

**Ethical Considerations:**

No, there are no or only very minor ethics concerns

**Final Justification:**

Thanks for your reply, your reply answered most of my questions and I improved my score.

**Limitations Weaknesses:**

1. The dataset is based on a specific legal system and amendment type, and its generalization ability across legal jurisdictions or different countries/regions needs to be further verified.

2. The construction of legal amendment scenarios, data collation and label design require a lot of legal expertise and manual proofreading, and the subsequent expansion and maintenance costs are high.

3. Although the architectural impact is revealed, there is still room for improvement in the explanation and interpretability tools of how the internal mechanism of the model adapts to the amendment.

4. The interpretation and application of legal provisions are inherently subjective and ambiguous. Although the deformation test reduces the reliance on labels, the interpretation of the results may still be affected by subjective factors.

**Strengths Contributions:**

1. It clearly focuses on the scenario of legal article revision, which meets the main challenges faced by real legal reasoning and has high application value.

2. It covers 31 types of fine-grained legal revisions, which can comprehensively evaluate the adaptability of the model to changes in different legal articles and is suitable for in-depth analysis.

3. It conducts a comprehensive empirical analysis of five SOTA models, clearly reveals the impact of model structure and design on adaptability, and proposes improvement directions, which is highly instructive.

---

> ### Author Rebuttal · Authors · 2025-07-31
>
> We appreciate the reviewer’ insightful comments and suggestions. To facilitate clarity, we provide a point-by-point response corresponding to each comment.
>
> ### **Generalization**
>
> First, our method for statute decomposition and revision type definition is widely generalizable. Regardless of legal system, codified rules can be represented as propositional statements or structured clauses, which our decomposition strategy is built to handle. For example, the U.K.’s Sentencing Council specifies key elements and penalties for Attempted Murder, noting “...the offender had an intention to kill; accordingly an offender convicted of this offence will have demonstrated a high level of culpability…” with “maximum: life imprisonment; offence range: 3–40 years’ custody.” Using our approach (Fig. 1), this statute can be decomposed into core components—subject (offender), action (kill), subject condition (intention)—and legal consequences (sentence range: 3–40 years). Although our revision taxonomy was developed from the Chinese Criminal Code, many types like T6.1 (term-num-up) and T2.1 (action-exp-ee) apply directly to such UK statutes.
>
> Second, our metamorphic testing approach for assessing model adaptability to legal revisions is broadly applicable. For instance, in the U.S., complexities arise when both statutes and precedents change. The 2022 Supreme Court decision in Dobbs v. Jackson Women’s Health Organization overturned the 1973 Roe v. Wade precedent that recognized abortion as a constitutional right, shifting regulatory authority back to the states. While Roe invalidated state-level abortion bans, Dobbs allows states to criminalize abortion. Thus, a Texas doctor performing abortions in 2021 acted lawfully, but identical conduct in 2023 may lead to prosecution, reflecting the 2022 state legislation banning abortion.
>
> We can design metamorphic test cases that capture the interplay of statutory and precedent changes (as shown in the Table). By presenting identical facts with variations in time, region, or precedent, we evaluate whether models adjust their judgments accordingly, reflecting adaptive legal reasoning. For example, in the first two abortion cases with identical facts but differing in time and precedent, the model should classify the later case as a crime and the earlier one as not a crime.
>
> We will revise the manuscript to include a discussion of these points. We appreciate the reviewer’s suggestion to elaborate on this important issue.
>
> |Test ID|Year|Region|Fact| Applicable Statute/Precedent|Expected Prediction|Reasoning Change Point|
> |---|---|---|---|---|---|---|
> |1|2021|Texas|Abortion|Roev.Wade|Not a crime|Based on Roe, the federal constitutional right to privacy protects a woman’s decision to have an abortion; states cannot impose a criminal prohibition.|
> |2|2023|Texas|Abortion|State law+Dobbs|Crime| After Dobbs overturned Roe, abortion regulation authority returned to the states. Texas enacted laws banning abortion, now enforceable without Roe's limits. |
> |3|2021|California|Abortion|Roe+Statelaw|Not a crime| While Roe established a federal protection baseline, California also had explicit state laws affirming abortion rights, reinforcing the non-criminal status.|
>
> ### **Scalability of LawShift**
>
> Thank you for this valuable observation. While constructing legal amendment scenarios initially demands legal expertise and manual effort, the process can be made scalable and sustainable
>
> We can leverage regex or LLMs to enable semi-/fully-automatic data annotation. For example, when the statutory subject expands from “poisonous substances” to “poisonous or radioactive substances,” an LLM can detect the original span and replace it with a sampled entity (e.g., “radium”) from a predefined list. This supports factual diversity but may introduce span-matching errors. To improve reliability while limiting manual effort, we can adopt human verification or ensemble LLM voting for more robust span detection.
>
> As noted in line 193, we rely on legal experts to ensure that revision type annotations are both accurate and legally plausible, e.g., avoiding unrealistic edits such as revising a theft charge into a capital offense. Nonetheless, to scale the annotation of revision types, we may also adopt semi-automatic or automatic methods. We can first use tools to locate the revision span (e.g., the penalty clause), and then replace or insert content from a curated set of candidates (e.g., alternative penalties for theft). Subsequently, expert review or voting-based aggregation can be used to reduce technical errors. Note human verification is applied at critical points to ensure legal fidelity, balancing cost and quality. Thus, while legal expertise is necessary, our workflow can be designed for long-term extensibility with manageable overhead.
>
>
>
> ### **Model Interpretation**
>
> While further exploration of model interpretability would strengthen the work, our focus in this paper is on benchmarking and identifying the limitations of existing models when faced with statute revisions. We provide a preliminary demonstration to respond to the comments.
>
> We performed basic knowledge tracing [1][2] experiments on Lawformer (Transformer-based) and NeurJudge (neural-based) and evaluated them with the action component’s revision types.
>
> Our goal is to analyze how the internal representations of LJP models evolve when encountering statutory revisions, and to identify which layers or modules are most responsible for failures. We conduct layer-wise and module-level tracing on Lawformer and NeurJudge.
>
> For Lawformer, we extract hidden states from all transformer layers (L1-L12) for both original and revised clauses. For NeurJudge, as it doesn’t have transformer blocks, we trace and record three vectors: (1) the final hidden state of the fact encoder, (2) the final hidden state of the law encoder, and (3) the pre-softmax output of the MatchNet.
>
> Every layer of Lawformer or every major sub-module of NeurJudge encodes a compressed summary of the statute it has read. By extracting these vectors before and after we swap in the amended clause, we obtain two snapshots of the models’ internal picture of the law.
>
> Then we run the model twice. Once with the original clauses, once with the revised clause. For the two runs of experiments, we trace and compute:
>
> $ \Delta^{k} = \left\lVert v^k_{orig}  -  v^k_{rev} \right\rVert_{2} $
>
>
> where k indexes layers for Lawformer and the three vectors for NeurJudge.
>
> Large gap means the semantics stored in layer k changes a lot when the clause changes. That layer is therefore sensitive to the amendment. A small gap shows that the layer is largely oblivious to clause changes. By scanning through all k, we learn where in the network the new legal knowledge is (or is not) understood.
>
> The results are listed in the following tables.
>
> | Layer  | $\Delta^{k}$ |
> |-----|------|
> |1|0.2791|
> |2|0.2434|
> |3|0.0710|
> |4|0.0305|
> |5|0.0245|
> |6|0.0175|
> |7|0.0113|
> |8|0.0097|
> |9|0.0104|
> |10|0.0109|
> |11|0.0102|
> |12|0.9911|
>
> | Module | $\Delta^{k}$|
> |-----|------|
> | fact encoder| 0.0 |
> | law encoder | 0.8915|
> |  match net | 0.0 |
>
>
> - Findings for the action component
>
> For Lawformer, the strongest perturbation occurs in encoder block 12. In the variance-normalized comparison of average hidden-state distances across all transformer blocks, block 12 shows the largest mean change, indicating that this block is the point most sensitive to action-clause revisions.
>
> For NeurJudge, the largest shift appears in the law encoder final state, thus the GRU law embeddings is the bottleneck through which action revisions must be absorbed.
>
> These tracing results provide a concrete roadmap for our future work on how to adapting to revisions. For Lawformer, the pronounced sensitivity of encoder block 12 suggests that subsequent research should target this block with interventions that specialise its feed-forward space for revised action clauses. For NeurJudge, the saliency peak in the law-encoder state indicates that upcoming efforts should explore representation-level patches or lightweight gating mechanisms that operate directly on that vector. By focusing on these empirically located memory slots, future editing/updating strategies can address statutory action revisions with maximal impact and minimal collateral drift.
>
> [1] Guidotti R, Monreale A, Ruggieri S, et al. A survey of methods for explaining black box models. ACM computing surveys, 2018, 51(5): 1-42.
>
> [2] Zhang Y, Tiňo P, Leonardis A, et al. A survey on neural network interpretability. IEEE transactions on emerging topics in computational intelligence, 2021, 5(5): 726-742.

---

> > ### Author Response · Authors · 2025-08-08
> > **Any outstanding questions or concerns?**
> >
> > Dear Reviewr xzpX,
> >
> > Since the author-reviewer discussion period is coming to a close, we would like to check with you once again to see if our responses satisfactorily address your concerns. If so, we appreciate a brief confirmation. If not, we would like to make use of the remaining time to answer any questions you may still have after reading our response. We are committed to making sure that you are completely satisfied with our responses.
> >
> > Best, Authors

---

> ### Author Response · Authors · 2025-08-06
> **Completely satisfied with our responses?**
>
> We sincerely appreciate the reviewer’s comments. Did our responses satisfactorily address all of your questions and comments? We are very eager to make sure that every concern and comment you may have is addressed. If everything looks good to you, please give us a confirmation. If any questions remain or further clarification is needed, we would be grateful for the opportunity to elaborate. We truly value your feedback and welcome further discussion and engagement.

---

### Official Review · Reviewer_7e8S · 2025-07-03

**Rating:** 4
**Confidence:** 3

**Summary:**

The authors propose a benchmark specifically designed to evaluate Legal Judgment Prediction models under statutory revisions. By modeling 31 types of fine-grained legal changes and using metamorphic testing, the benchmark systematically assesses whether models can adapt to evolving statutes. The authors show that existing SOTA models struggle significantly in this setting.

**Additional Feedback:**

Some references are missing and some typos exists.

**Dataset Code Accessibility:**

Yes

**Dataset Code Comments:**

Code and dataset is provided in the paper.

**Ethical Considerations:**

No, there are no or only very minor ethics concerns

**Final Justification:**

The authors have made a commendable effort in addressing all of the concerns and follow-up questions I raised. As a result, I am revising my score upwards.

**Limitations Weaknesses:**

The paper currently focus only on Chinese Criminal Code so generalization across other legal systems is a question here?
So keen to know more about the authors insights about the generalizability.

Wondering about the possible broader evaluation? How about testing with adaptable versions of the sotas and how well it can perform will be more interesting to gain better insights.

The selection of five baselines need more supporting statements, I am not sure how recent those selected ones are? In the related work there was discussion about LLM based approaches, however none was used in the evaluation for comparison.

Since the fine-grained level of statutory revisions can be complex, in terms of experts involvement, how challenging it can be in a more higher scale of data (more number from 31 and include data from more years)?

**Strengths Contributions:**

The authors addresses a highly relevant gap in legal AI by handling dynamic statutory changes, which is relevant for the dynamic real-world use.

The proposed dataset benchmark involve guidance from legal experts that ensure the validity of the benchmark dataset. It also attempted to analyse a fine-grained statutory revision types with the intention to capture the complexity.

The authors provide a detailed analysis of model performance across different revision types, offering insights for model design.

---

> ### Author Rebuttal · Authors · 2025-07-31
>
> We appreciate the reviewer's constructive evaluation of our work. We address each comment individually as follows.
>
> ### **Generalization**
>
> First, our method for statute decomposition and revision type definition is widely generalizable. Regardless of legal system, codified rules can be represented as propositional statements or structured clauses, which our decomposition strategy is built to handle. For example, the U.K.’s Sentencing Council specifies key elements and penalties for Attempted Murder, noting “...the offender had an intention to kill; accordingly an offender convicted of this offence will have demonstrated a high level of culpability…” with “maximum: life imprisonment; offence range: 3–40 years’ custody.” Using our approach (Fig. 1), this statute can be decomposed into core components—subject (offender), action (kill), subject condition (intention)—and legal consequences (sentence range: 3–40 years). Although our revision taxonomy was developed from the Chinese Criminal Code, many types like T6.1 (term-num-up) and T2.1 (action-exp-ee) apply directly to such UK statutes.
>
> Second, our metamorphic testing approach for assessing model adaptability to legal revisions is broadly applicable. For instance, in the U.S., complexities arise when both statutes and precedents change. The 2022 Supreme Court decision in Dobbs v. Jackson Women’s Health Organization overturned the 1973 Roe v. Wade precedent that recognized abortion as a constitutional right, shifting regulatory authority back to the states. While Roe invalidated state-level abortion bans, Dobbs allows states to criminalize abortion. Thus, a Texas doctor performing abortions in 2021 acted lawfully, but identical conduct in 2023 may lead to prosecution, reflecting the 2022 state legislation banning abortion.
>
> We can design metamorphic test cases that capture the interplay of statutory and precedent changes (as shown in the Table). By presenting identical facts with variations in time, region, or precedent, we evaluate whether models adjust their judgments accordingly, reflecting adaptive legal reasoning. For example, in the first two abortion cases with identical facts but differing in time and precedent, the model should classify the later case as a crime and the earlier one as not a crime.
> We will revise the manuscript to include a discussion of these points. We appreciate the reviewer’s suggestion to elaborate on this important issue.
>
> |Test ID|Year|Region|Fact| Applicable Statute/Precedent|Expected Prediction|Reasoning Change Point|
> |---|---|---|---|---|---|---|
> |1|2021|Texas|Abortion|Roev.Wade|Not a crime|Based on Roe, the federal constitutional right to privacy protects a woman’s decision to have an abortion; states cannot impose a criminal prohibition.|
> |2|2023|Texas|Abortion|State law+Dobbs|Crime| After Dobbs overturned Roe, abortion regulation authority returned to the states. Texas enacted laws banning abortion, now enforceable without Roe's limits. |
> |3|2021|California|Abortion|Roe+Statelaw|Not a crime| While Roe established a federal protection baseline, California also had explicit state laws affirming abortion rights, reinforcing the non-criminal status.|
>
> ### **Broader Evaluation**
>
> We appreciate the reviewer’s suggestion and we agree that developing a new model would be valuable. Our main contribution is introducing LawShift, the first benchmark dataset, which we view as a foundational step for this research area. While model design tailored to this task holds promise, it deserves a dedicated study beyond this initial benchmark paper. Nevertheless, we conducted preliminary experiments in response to the comment.
>
> We integrate ROME [1], a SOTA knowledge update method, into Lawformer and NeurJudge to evaluate its effectiveness in addressing legal revision challenges.
>
> We integrate ROME into Lawformer to inject revised legal knowledge at specific model layers. First, we identify which encoder layer is most sensitive to a specific law component (e.g., subject, action, term) by measuring how much its representation shifts after statute revisions. Then, following ROME, we apply a targeted rank-one update to the feed-forward weights of that layer. This update ensures that the model produces the correct post-revision prediction while preserving performance on pre-revision and unrelated cases, allowing efficient and localized legal knowledge editing without retraining the entire model. We apply the same approach to incorporate ROME into NeurJudge. Due to the rebuttal's length constraints, we have omitted the detailed experimental steps but would be happy to provide them during the author-reviewer discussion.
>
> | Type  | Lawformer (Ori) | Lawformer+ROME | NeurJudge (Ori) | NeurJudge+ROME|
> |-----|------|--------|------|--------|
> |T1.1|89.27|90.42|88.51|87.74|
> |T1.2|88.51|88.89|86.21|86.97|
> |T1.3|89.66|88.89|87.74|87.74|
> |T1.4|12.64|24.52|12.64|14.56|
> |T1.5|12.26|22.99|12.64|15.71|
> |T1.6|7.66|21.07|8.05|8.81|
> |T2.1|99.02|99.02|99.20|98.04|
> |T2.2|0.00|19.61|69.61|70.59|
> |T2.3|7.84|12.75|78.43|78.43|
> |T2.4|97.06|94.12|18.63|21.57|
> |T2.5|6.86|18.63|16.67|22.55|
> |T2.6|4.90|16.67|9.80|12.75|
> |T3.1|97.60|98.04|96.30|98.26|
> |T3.2|94.77|94.99|85.84|89.32|
> |T3.3|94.77|94.34|87.36|90.41|
> |T3.4|9.37|19.83|13.72|15.03|
> |T3.5|10.02|20.70|13.29|15.03|
> |T3.6|9.37|20.04|9.80|10.89|
> |T4.1|4.67|14.00|3.33|8.67|
> |T4.2|4.67|15.33|2.00|8.67|
> |T4.3|4.67|13.33|2.00|8.00|
> |T4.4|93.33|94.00|98.00|96.00|
> |T4.5|92.67|91.33|98.67|98.00|
> |T4.6|3.18|8.07|6.11|7.33|
> |T4.7|91.95|93.96|97.99|96.64|
> |T5.1|13.19|18.72|12.34|13.40|
> |T5.2|81.74|86.96|80.00|81.74|
> |T6.1|9.77|16.28|0.00|8.37|
> |T6.2|42.79|52.56|69.77|70.70|
> |T6.3|7.91|12.56|4.65|8.37|
> |T6.4|7.07|11.11|0.00|3.03|
>
> - Result and analysis
>
> Introducing ROME significantly boosts both backbones’ ability to track statutory changes. For Lawformer, previously challenging revisions improve notably: subject reallocation reduction T1.4 rises from 12.64% to 24.52%, explicit subject removal T1.5 from 12.26% to 22.99%, and implicit reduction T1.6 nearly triples to 21.07%. Action-scope edits follow a similar trend: implicit expansion T2.2 jumps from 0% to 19.61%, explicit removal T2.5 from 6.86% to 18.63%, and implicit removal T2.6 from 4.90% to 16.67%. Sentencing shifts also gain, with term-up T6.1 moving from 9.77% to 16.28% and extreme-term insertion T6.3 from 7.91% to 12.56%. Tasks requiring unchanged predictions (T2.1, T3.1, T4.4, T4.5) maintain near-perfect scores, confirming the edits’ locality and preservation of prior knowledge.
>
> NeurJudge shows smaller but consistent gains: explicit action removal T2.5 rises from 16.67% to 22.55%, object removal T3.5 from 13.29% to 15.03%, and objective-condition expansion T4.1 from 3.33% to 8.67%. Numerical sentencing revisions improve as well: T6.1 climbs from 0% to 8.37%, and extreme-term elimination T6.4 reaches 3.03%.
>
> Despite improvements, LawShift remains challenging. Revisions relying on implicit rephrasing (T1.6, T2.6, T3.6) remain around or below 20%. Complex multi-step event reallocations are challenging for NeurJudge, and sentencing changes involving scale inversion or extreme penalties still have low success rates. These persistent errors highlight two key obstacles: many amendments shift semantics (reasoning) without explicit lexical cues, limiting token-level edits; and several revision types require richer event-centric abstraction, aligning dispersed factual mentions with revised statutes. Thus, LawShift exposes subtle semantic shifts demanding deep structural legal understanding, and while lightweight edits help, they only partially address this complexity.
>
> [1] Meng, Kevin, et al. Locating and editing factual associations in gpt. NIPS 35 (2022): 17359-17372.
>
> ### **Clarifying Baselines**
>
> The five baselines we selected represent state-of-the-art and widely recognized approaches in legal judgment prediction (LJP), spanning traditional neural models, pre-trained legal-specific models, and LLM-based methods. As outlined in Appendix C (line 582), these models cover key paradigms in LJP. Notably, three of these (D-LADAN, NeurJudge, Qwen2.5) are recent models (2024–2025), ensuring our evaluation reflects both foundational and cutting-edge work. For LLM-based approaches, we selected Qwen2.5-7B due to its top performance on legal benchmarks such as LawBench (Fei et al., 2023), which we cite in the related work (lines 76–77). This inclusion ensures our evaluation captures not only traditional and PLM-based methods but also the latest advances in LLM-based legal reasoning.
>
> ### **Scalability of LawShift**
>
> The process can be made scalable and sustainable
>
> We can leverage regex or LLMs to enable semi-/fully-automatic data annotation. For instance, when the statutory subject expands from “poisonous substances” to “poisonous or radioactive substances,” an LLM can detect the original span and replace it with a sampled entity (e.g., “radium”) from a predefined list. This supports factual diversity but may introduce span-matching errors. To improve reliability while limiting manual effort, we can adopt human verification or ensemble LLM voting for more robust span detection.
>
> As noted in line 193, we rely on legal experts to ensure that revision type annotations are legally plausible, e.g., avoiding unrealistic edits such as revising a theft charge into a capital offense. However, to scale the annotation of revision types, we may also adopt semi-automatic or automatic methods. We can first use tools to locate the revision span (e.g., the penalty clause), and then replace or insert content from a curated set of candidates (e.g., alternative penalties for theft). Subsequently, expert review or voting-based aggregation can be used to reduce technical errors. Note human verification is applied at critical points to ensure legal fidelity, balancing cost and quality. Thus, while legal expertise is necessary, our workflow can be designed for long-term extensibility with manageable overhead.

---

> > ### Comment · Reviewer_7e8S · 2025-08-04
> > **Response to author's rebuttal**
> >
> > I appreciate the author's effort and the work put into this rebuttal. However, I would like some further clarification on a couple of points:
> >
> > Generalizability: The examples presented are interesting, particularly those showcasing generalizability using cases from two countries. Could you provide additional insights into the test case design? Specifically, I would appreciate more information on how the countries are chosen for these cases. Also, it would be valuable to understand what extreme failure cases might arise if specific laws from certain countries are selected.
> >
> > Could you please elaborate on the results of T2.2 and T4.1 to T4.3? There appears to be a significant drop in performance on the dataset in these cases, while T2.1 and similar cases show better performance. Any insights into what factors could contribute to such a large discrepancy?
> >
> > Clarification on “LawShift” Statement: When the authors mention that “LawShift exposes subtle semantic shifts demanding deep structural legal understanding, and while lightweight edits help, they only partially address this complexity,” what do authors mean? Is it suggesting that a model needs to be trained from scratch, or is there a need for a more sophisticated model than the one currently used?

---

> > > ### Author Response · Authors · 2025-08-05
> > > **Response to Reviewer’s Follow-up Comments (1/2)**
> > >
> > > Thank you for your encouraging feedback and thoughtful questions. We address these points in turn.
> > > 1. **Generalizability**
> > >
> > > We specifically selected countries governed by common law systems (i.e., the UK and the US) to contrast with civil law jurisdictions such as the Chinese legal framework used in our paper, in order to better demonstrate the methodology’s generalizability across different legal frameworks.
> > >
> > > We appreciate the reviewer’s suggestion to explore potential extreme failure cases. We argue that our methodology is broadly applicable across most jurisdictions, as the majority adhere to civil law, common law, or a combination of both. Nonetheless, there exist certain extreme cases where our methodology may not be directly applicable.
> > >
> > > One example lies in jurisdictions governed by Islamic law. In some Islamic legal systems, the law is derived primarily from the Quran and Hadith, and statutory provisions tend to be abstract, open-textured, and lacking standardized syntactic structures. These provisions do not follow the typical condition-consequence pattern found in civil or common law, making them incompatible with our decomposition strategy. Moreover, legal revisions are rarely enacted through explicit textual amendments. Instead, changes arise through religious interpretations (e.g., fatwas), without producing revisable legislative artifacts. This makes it infeasible to construct structured revision test cases in a way compatible with our framework. Furthermore, in certain regions, such as rural Islamic communities, legal norms may exist primarily in the form of oral customary law rather than codified statutes. In such settings, the absence of formal legal texts prevents the application of our revision-based methodology.
> > >
> > > In jurisdictions lacking codified, structured, or explicitly revised legal texts, our methodology may not be directly applicable without adaptation. We appreciate the reviewer’s insightful comment about extreme failure cases.
> > >
> > > 2. **Elaborations on the results of T2.2 and T4.1 to T4.3**
> > >
> > > Revisions in T4.1 to T4.3 are expected to induce changes in judgment predictions, while the semantically similar revision in T4.4 is expected to preserve the original prediction. Most LJP models perform reliably when predictions are expected to remain consistent but struggle to adjust when revised statutes require different outcomes (lines 279-282). LJP models tend to over-rely on statistical co-occurrence patterns between case facts and legal judgments learned during training, rather than explicitly modeling the legal reasoning relationship between them. As a result, even when the statute is revised (e.g., T4.1 to T4.3), the model may continue to make predictions based on pre-existing patterns, rather than re-evaluating the legal consequences under the updated law. This suggests that the model fails to treat statutory changes as primary drivers of legal decisions.
> > >
> > > As for the significant performance drop on T2.2, which happened to the Lawformer model, it represents a notable exception. Relevant statistics are listed in the following table (These data can also be found in Table 10 of our original manuscript).
> > > |**Type**|**Pass Rate Before Revision**|**Pass Rate After Revision (Ori)** |**Pass Rate After Revision+ROME**|
> > > |--------|--------|---------|------|
> > > |T2.1| 100.0|99.02|99.02|
> > > |T2.2|100.0|0.00|19.61|
> > > |T2.3|100.0|7.84|21.07|
> > >
> > > Notably, T2.2 and T2.3 are frequently misclassified under a confusing law article, which is a pattern not observed before revision. Specifically, Lawformer exhibits a tendency to misclassify harboring as drug possession. A possible explanation is that, following the revision, the case descriptions include phrases such as “harboring individuals for drug transactions”, which may introduce semantic overlap with offenses related to drug possession. This semantic proximity appears to confuse the model, leading to incorrect predictions. The pattern suggests that on T2.2 Lawformer relies on surface-level keyword associations rather than a nuanced understanding of legal semantics and their contextual application.

---

> > > > ### Author Response · Authors · 2025-08-05
> > > > **Response to Reviewer’s Follow-up Comments (2/2)**
> > > >
> > > > 3. **Clarification on “LawShift” Statement**
> > > >
> > > > We appreciate the reviewer’s question. Our response stresses the need for a more sophisticated model, because simply retraining is often impractical for two key reasons. First, full retraining is prohibitively expensive, both computationally and financially, given that LJP models typically depend on large-scale legal corpora. Second, when statutes are amended there is rarely a sufficient volume of post-amendment cases to support effective retraining. Consequently, we require an architecture that can flexibly inject updated legal knowledge without rebuilding the entire model from scratch.
> > > >
> > > > A more sophisticated model is necessary. Even seemingly minor statutory revisions can substantially alter the legal reasoning process, which often involves multi-step inference chains (e.g., evidence → sub-conclusion → final legal decision). This depth of reasoning far exceeds the capabilities of current knowledge editing techniques, which typically focus on straightforward logical updates, such as changing the identity of the President of the United States from person A to person B. In contrast, statutory changes require models not only to detect fine-grained semantic shifts but also to reconstruct new logical pathways that reflect updated statutes. Therefore, we argue that addressing statute revision challenges calls for a dedicated model.

---

> > > > > ### Comment · Reviewer_7e8S · 2025-08-05
> > > > > **Reply to authors**
> > > > >
> > > > > Thank you for your efforts in providing a better clarity on the follow-up questions, much appreciated!.
> > > > > I will revise my score accordingly.

---

### Official Review · Reviewer_y3ZJ · 2025-07-04

**Rating:** 3
**Confidence:** 4

**Summary:**

This paper introduces LawShift, a new benchmark dataset for evaluating how Legal Judgment Prediction (LJP) models perform when faced with changes in underlying laws (statutory revisions). LawShift is built around 31 fine-grained types of legal changes. The paper presents an evaluation of five LJP models on LawShift, highlighting their general struggles in adapting to these legal shifts.

**Dataset Code Accessibility:**

Yes

**Ethical Considerations:**

Yes, there are significant ethics concerns that require review by an ethics expert

**Final Justification:**

Thanks to the authors for providing extensive additional experiments and important clarifications. Based on the current version, the paper requires further revisions before it is ready for acceptance.

**Limitations Weaknesses:**

While the authors claim the methodology is generalizable, this is not substantiated. Could the author elaborate on how the 31 revision types would translate to other legal systems, particularly common law systems (e.g., U.S., U.K.) where precedent plays a larger role? A more robust discussion on the challenges and a potential roadmap for adaptation would be needed to support this claim.

The paper shows that existing models fail on this new task. The contribution would be stronger if the authors also provide a new model/method that is designed to handle this. This would show how hard the task truly is and would point future research in a more promising direction.

The analysis of why models fail could be deeper. For example, the paper notes that action revisions are the most challenging, attributing it to a need for "deeper contextual comprehension." This explanation is vague. Is it possible to provide an in-depth analysis or a more concrete hypothesis for this observation? Are there specific linguistic or structural properties of how "actions" are described in legal text that make them particularly difficult for models to process compared to "subjects" or "objects"?

**Strengths Contributions:**

The paper highlights an important and often overlooked issue in the LJP domain: the need for models to adapt to evolving legal standards.

The use of metamorphic testing is a clever and appropriate strategy for this specific problem.

The paper is clearly written and provides significant detail, especially in the appendices, regarding the construction of the dataset and the experimental setup.

---

> ### Author Rebuttal · Authors · 2025-07-31
>
> We sincerely thank the reviewer for the insightful and constructive comments. Below, we address each point in detail.
>
> ### **Generalization**
>
> First, our method for statute decomposition and revision type definition is widely generalizable. Regardless of legal system, codified rules can be represented as propositional statements or structured clauses, which our decomposition strategy is built to handle. For example, the U.K.’s Sentencing Council specifies key elements and penalties for Attempted Murder, noting “...the offender had an intention to kill; accordingly an offender convicted of this offence will have demonstrated a high level of culpability…” with “maximum: life imprisonment; offence range: 3–40 years’ custody.” Using our approach (Fig. 1), this statute can be decomposed into core components—subject (offender), action (kill), subject condition (intention)—and legal consequences (sentence range: 3–40 years). Although our revision taxonomy was developed from the Chinese Criminal Code, many types like T6.1 (term-num-up) and T2.1 (action-exp-ee) apply directly to such UK statutes.
>
> Second, our metamorphic testing approach for assessing model adaptability to legal revisions is broadly applicable. For instance, in the U.S., complexities arise when both statutes and precedents change. The 2022 Supreme Court decision in Dobbs v. Jackson Women’s Health Organization overturned the 1973 Roe v. Wade precedent that recognized abortion as a constitutional right, shifting regulatory authority back to the states. While Roe invalidated state-level abortion bans, Dobbs allows states to criminalize abortion. Thus, a Texas doctor performing abortions in 2021 acted lawfully, but identical conduct in 2023 may lead to prosecution, reflecting the 2022 state legislation banning abortion.
>
> We can design metamorphic test cases that capture the interplay of statutory and precedent changes (as shown in the Table). By presenting identical facts with variations in time, region, or precedent, we evaluate whether models adjust their judgments accordingly, reflecting adaptive legal reasoning. For example, in the first two abortion cases with identical facts but differing in time and precedent, the model should classify the later case as a crime and the earlier one as not a crime.
> We will revise the manuscript to include a discussion of these points. We appreciate the reviewer’s suggestion to elaborate on this important issue.
>
> |Test ID|Year|Region|Fact| Applicable Statute/Precedent|Expected Prediction|Reasoning Change Point|
> |---|---|---|---|---|---|---|
> |1|2021|Texas|Abortion|Roev.Wade|Not a crime|Based on Roe, the federal constitutional right to privacy protects a woman’s decision to have an abortion; states cannot impose a criminal prohibition.|
> |2|2023|Texas|Abortion|State law+Dobbs|Crime| After Dobbs overturned Roe, abortion regulation authority returned to the states. Texas enacted laws banning abortion, now enforceable without Roe's limits. |
> |3|2021|California|Abortion|Roe+Statelaw|Not a crime| While Roe established a federal protection baseline, California also had explicit state laws affirming abortion rights, reinforcing the non-criminal status.|
>
> ### **Broader Evaluation**
>
> We appreciate the reviewer’s suggestion and we agree that developing a new model would be valuable. Our main contribution is introducing LawShift, the first benchmark dataset, which we view as a foundational step for this research area. While model design tailored to this task holds promise, it deserves a dedicated study beyond this initial benchmark paper. Nevertheless, we conducted preliminary experiments in response to the comment.
>
> We integrate ROME [1], a SOTA knowledge update method, into Lawformer and NeurJudge to evaluate its effectiveness in addressing legal revision challenges.
>
> We integrate ROME into Lawformer to inject revised legal knowledge at specific model layers. First, we identify which encoder layer is most sensitive to a specific law component (e.g., subject, action, term) by measuring how much its representation shifts after statute revisions. Then, following ROME, we apply a targeted rank-one update to the feed-forward weights of that layer. This update ensures that the model produces the correct post-revision prediction while preserving performance on pre-revision and unrelated cases, allowing efficient and localized legal knowledge editing without retraining the entire model. We apply the same approach to incorporate ROME into NeurJudge. Due to the rebuttal's length constraints, we have omitted the detailed experimental steps but would be happy to provide them during the author-reviewer discussion.
>
> | Type  | Lawformer (Ori) | Lawformer+ROME | NeurJudge (Ori) | NeurJudge+ROME|
> |-----|------|--------|------|--------|
> |T1.1|89.27|90.42|88.51|87.74|
> |T1.2|88.51|88.89|86.21|86.97|
> |T1.3|89.66|88.89|87.74|87.74|
> |T1.4|12.64|24.52|12.64|14.56|
> |T1.5|12.26|22.99|12.64|15.71|
> |T1.6|7.66|21.07|8.05|8.81|
> |T2.1|99.02|99.02|99.20|98.04|
> |T2.2|0.00|19.61|69.61|70.59|
> |T2.3|7.84|12.75|78.43|78.43|
> |T2.4|97.06|94.12|18.63|21.57|
> |T2.5|6.86|18.63|16.67|22.55|
> |T2.6|4.90|16.67|9.80|12.75|
> |T3.1|97.60|98.04|96.30|98.26|
> |T3.2|94.77|94.99|85.84|89.32|
> |T3.3|94.77|94.34|87.36|90.41|
> |T3.4|9.37|19.83|13.72|15.03|
> |T3.5|10.02|20.70|13.29|15.03|
> |T3.6|9.37|20.04|9.80|10.89|
> |T4.1|4.67|14.00|3.33|8.67|
> |T4.2|4.67|15.33|2.00|8.67|
> |T4.3|4.67|13.33|2.00|8.00|
> |T4.4|93.33|94.00|98.00|96.00|
> |T4.5|92.67|91.33|98.67|98.00|
> |T4.6|3.18|8.07|6.11|7.33|
> |T4.7|91.95|93.96|97.99|96.64|
> |T5.1|13.19|18.72|12.34|13.40|
> |T5.2|81.74|86.96|80.00|81.74|
> |T6.1|9.77|16.28|0.00|8.37|
> |T6.2|42.79|52.56|69.77|70.70|
> |T6.3|7.91|12.56|4.65|8.37|
> |T6.4|7.07|11.11|0.00|3.03|
>
> -  **Result and analysis**
>
> Introducing ROME significantly boosts both backbones’ ability to track statutory changes. For Lawformer, previously challenging revisions improve notably: subject reallocation reduction T1.4 rises from 12.64% to 24.52%, explicit subject removal T1.5 from 12.26% to 22.99%, and implicit reduction T1.6 nearly triples to 21.07%. Action-scope edits follow a similar trend: implicit expansion T2.2 jumps from 0% to 19.61%, explicit removal T2.5 from 6.86% to 18.63%, and implicit removal T2.6 from 4.90% to 16.67%. Sentencing shifts also gain, with term-up T6.1 moving from 9.77% to 16.28% and extreme-term insertion T6.3 from 7.91% to 12.56%. Tasks requiring unchanged predictions (T2.1, T3.1, T4.4, T4.5) maintain near-perfect scores, confirming the edits’ locality and preservation of prior knowledge.
>
> NeurJudge shows smaller but consistent gains: explicit action removal T2.5 rises from 16.67% to 22.55%, object removal T3.5 from 13.29% to 15.03%, and objective-condition expansion T4.1 from 3.33% to 8.67%. Numerical sentencing revisions improve as well: T6.1 climbs from 0% to 8.37%, and extreme-term elimination T6.4 reaches 3.03%.
>
> Despite improvements, LawShift remains challenging. Revisions relying on implicit rephrasing (T1.6, T2.6, T3.6) remain around or below 20%. Complex multi-step event reallocations are challenging for NeurJudge, and sentencing changes involving scale inversion or extreme penalties still have low success rates. These persistent errors highlight two key obstacles: many amendments shift semantics (reasoning) without explicit lexical cues, limiting token-level edits; and several revision types require richer event-centric abstraction, aligning dispersed factual mentions with revised statutes. Thus, LawShift exposes subtle semantic shifts demanding deep structural legal understanding, and while lightweight edits help, they only partially address this complexity.
>
> [1] Meng, Kevin, et al. Locating and editing factual associations in gpt. NIPS 35 (2022): 17359-17372.
>
> ### **Deeper Analysis of Model Failures**
>
> Actions are typically expressed through complex event descriptions, often involving multi-step process, causative verbs, and temporal or conditional modifiers (e.g., discharge pollutants causing harm, or breaches the dam resulting in flooding). Such event-centered constructs require understanding causality, temporal sequence, identifying cross-sentence composition, and implicit conditions, which is more demanding than recognizing the relatively static and well-defined nominal structures of subjects or objects.
>
> From a linguistics perspective, prior studies have shown that verbs and event structures are semantically richer and syntactically more variable than nouns, often requiring compositional interpretation across clauses [1, 2]. Vendler [1] argues verbs encode temporal boundaries and event dynamics, which inherently increases their interpretive complexity. Pustejovsky [2] further argues that verbs require a multi-layered semantic representation (e.g., event structure, argument structure, etc.), making the understanding of actions dependent on both context and sub-event reasoning rather than on isolated lexical cues. (i.e., what we mean by requiring deeper contextual understanding).
>
> In our experiments, models based on word2vec+deep learning baselines fail because their embedding representations lack the capacity to encode complex event semantics and primarily rely on co-occurrence patterns. The pretrained Lawformer model, while leveraging transformer-based contextual embeddings, is still limited by token-level attention without explicit event decomposition, bringing challenges when the action semantics are paraphrased or restructured.
>
> In our final manuscript, we will expand the failure analysis section to include these insights.
>
> [1] Vendler, Zeno. Linguistics in philosophy. Cornell University Press, 1967.
>
> [2] Pustejovsky, James. The generative lexicon. MIT press, 1998.

---

> > ### Comment · Reviewer_y3ZJ · 2025-08-06
> >
> > Thanks for the detailed response. I have a few additional points that I would appreciate some clarification on.
> >
> > For the additional ROME-based experiments demonstrating improvements on several revision types. However, despite gains on tasks such as T1.6, T4.6, T6.1, T6.3, and T6.4, the accuracy remains low (mostly below 20%), indicating that ROME’s effectiveness on these types is still limited. Could the authors elaborate on why ROME struggles with specific types?
> >
> > The linguistic discussion on event structure and verb complexity is helpful. To further support the claim that deeper contextual understanding is needed, it would be valuable to see concrete examples of failure cases from the dataset. For instance, are there specific linguistic patterns or constructions, such as causatives or conditionals, that consistently challenge the models?

---

> > > ### Author Response · Authors · 2025-08-07
> > > **Response to the reviewer’s follow-up comments (1/2)**
> > >
> > > **1. Elaborations on why ROME struggles with specific types**
> > >
> > > We thank the reviewer for this insightful observation. While ROME yields improvements on several revision types (e.g., T1.6, T4.6, T6.1, T6.3, and T6.4), the overall accuracy remains low.
> > >
> > > This limitation primarily stems from the nature of the statute revisions in LJP, which often involves complex semantic shifts and changes in legal reasoning. ROME and similar knowledge editing methods are typically optimized for injecting or updating discrete factual knowledge (e.g., subject–relation–object triples, Trump-Is-President). In other words, these methods typically focus on updating straightforward logics, such as changing the president of the US from person A to person B. However, legal statute revisions require more complex structural inference, not merely recognizing discrete factual knowledge, but reasoning over legal definitions, synthesizing evidence scattered across multiple sentences, and understanding underlying legal relationships.
> > >
> > > A representative case is revision type T1.6, which requires identifying subjects within an implicitly narrowed legal scope. For example, the original formulation “personnel of state-owned companies, enterprises, or other state-owned entities” is implicitly narrowed to “individuals in charge of state-owned companies”. In this context, the term “subjects” refers not to all personnel but specifically to those exercising managerial or controlling authority within the organization.
> > >
> > > While knowledge editing methods like ROME can update surface-level knowledge, e.g., updating semantics of “personnel of state-owned entities” to “individuals in charge of state-owned companies”, they fall short when deeper legal understanding is required. To make accurate predictions, LJP models must go beyond semantics to infer the legal definition of control and responsibility, evaluate the institutional relationship between individuals and entities, and assess the implications of managerial authority. This reasoning often spans multiple sentences and involves latent structural inferences, such as determining whether the evidence supports that the defendant qualifies as a controlling individual under relevant statutes.
> > >
> > > Such knowledge is rarely encoded in the form of explicit (subject–relation–object) triples or the localized causal chains that ROME typically edits. As a result, ROME struggles to revise model behavior when the update requires structural inferences beyond discrete factual knowledge. This illustrates a broader limitation of existing knowledge editors like ROME: they are effective for atomic fact corrections but insufficient for scenarios requiring multi-step reasoning.
> > >
> > > Another reason why ROME performs poorly on certain revision types is that the underlying LJP models themselves already struggle with these types. For example, T6.1, T6.3, and T6.4 all involve predicting the term of penalty—a particularly challenging task even for SOTA LJP models. Since the LJP model’s performance is weak in these areas, the improvements that ROME can bring are inherently limited. Similarly, revision types such as T1.6 and T4.6 are designed to trigger changes in the judgment outcomes, yet existing LJP models fail to capture these shifts accurately (lines 279-282). As a result, ROME can only offer modest gains.
> > >
> > > We hope this explanation clarifies why ROME struggles with these more complex revision types, and we appreciate the reviewer’s suggestion to further elaborate on this point.

---

> > > ### Author Response · Authors · 2025-08-07
> > > **Response to the reviewer’s follow-up comments (2/2)**
> > >
> > > **2. Failure cases**
> > >
> > > We appreciate the reviewer’s recommendation to provide concrete examples of failure cases observed in the dataset.
> > >
> > > A representative example appears in revision type T1.6 (identifying subjects in implicitly reduced scope), which requires the model to infer responsibility even when the subject is not explicitly framed as a legal actor. Consider the sentence: “The financial loss was caused by the [subject]’s unauthorized transfer of funds.” Here, the subject is not directly stated as liable, but the culpability is implicitly embedded in the causative passive construction (“was caused by”). However, the model often fails to make this attribution, treating the action as a neutral factual description rather than linking it to a legal consequence.
> > >
> > > A similar issue arises in T2.6, where the models’ failure to accurately identify the causative nature of an event critically impairs their ability to determine culpability. In this revision type, models are expected to distinguish between coercive behavior (“detaining others to inject or inhale drugs”, which should lead to a guilty verdict) and non-coercive actions (“harboring others who voluntarily inject or inhale drugs”, which should not). However, current models often overlook subtle but legally crucial indicators of coercion embedded in the event descriptions. This suggests that they conflate voluntary facilitation with active compulsion, revealing a lack of fine-grained semantic reasoning necessary to discern causative agency. As a result, they struggle to make legally sound distinctions in cases where the legal outcome hinges on nuanced interpretations of intent.
> > >
> > > Beyond surface-level event detection, models should be capable of interpreting such underlying “legal semantics” to capture the legal implications conveyed in natural language.
> > >
> > > Thank you for the helpful suggestion.

---

> > > > ### Comment · Reviewer_y3ZJ · 2025-08-07
> > > >
> > > > I thank the authors for their detailed responses and the inclusion of additional experiments. That said, several important points raised by other reviewers may benefit from further attention in a future revision.
> > > >
> > > > Limited generalization across legal systems: The dataset focuses exclusively on revisions to the Chinese Criminal Code. As noted by multiple reviewers, this raises important questions about the applicability and scalability of the proposed framework to other jurisdictions or types of legal amendments.
> > > >
> > > > Limited evaluation scope: Several reviewers suggested broader evaluation efforts, such as testing adaptable versions of recent state-of-the-art models, to better understand model limitations and generalizability.
> > > >
> > > > Given these considerations, I am inclined to maintain my original score.

---

> > ### Author Response · Authors · 2025-08-08
> > **Serious misunderstandings (1/2)**
> >
> > We thank the reviewer for your response. Since it looks like some serious misunderstanding exists, we think it’s crucial to clarify your misunderstanding by providing additional background information.
> >
> > **1. Limited generalization across legal systems**
> >
> > We provided examples in our response to you and other reviewers that demonstrate how our methodology can generalize across both civil and common law systems, such as by incorporating illustrative cases from the UK and the US. Reviewer 7e8S was already satisfied with our response on generalizability, saying that "the examples presented are interesting, particularly those showcasing generalizability using cases from two countries." If you haven't had a chance to read our response, we strongly encourage you to do so.
> >
> > We believe it'd be important for us to provide you with additional background on research in Legal AI/NLP, as we understand that it's easy for reviewers lacking background in this area to under-appreciate our work. First, this is not some generic machine learning dataset where one can simply ask some laymen (such as Amazon Mechanical Turkers) to provide the labels. Legal revisions require precise and legally valid annotations that only domain experts can provide. Without such domain-specific knowledge, the resulting data would lack the reliability and legal fidelity required for research use. Requiring a research team to possess legal expertise on multiple jurisdictions, possibly involving multiple natural languages, to produce multilingual legal annotations is simply not a realistic expectation. It is for this reason that there currently doesn't exist any legal corpora **with human-labeled annotations** that involve more than a single jurisdiction: researchers in this field all understand that is simply not a realistic expectation, and this is simply not the way how research in Legal AI/NLP works. (Note: while there are legal AI/NLP corpora that cover multiple jurisdictions and languages [1][2], they are all **raw, unannotated** corpora researchers downloaded from the Web and therefore can be assembled without legal expertise in multiple jurisdictions/languages.) And for this very same reason, we believe that when the reviewers commented on generalizability, they did not mean that we should show generalizability by annotating data in another jurisdiction using our framework, and that's why Reviewer 7e8S was satisfied when we provided examples that illustrate generalizability.
> >
> > Equally importantly, the fact that an legal AI/NLP corpus covers only one jurisdiction by no means undermines its usability and impact. For example, CAIL-2018 [3], which focuses on the Chinese jurisdiction, has been widely used in the field since its release and received 378 citations in the past seven years. As another example, ECtHR-2019 [4], which focuses on cases under the jurisdiction of the European Court of Human Rights, received 472 citations in the past six years and got 316 downloads in the last month on huggingface [5].
> >
> > We therefore have no reason to believe that our corpus in its current state is deficient to the point that it is not usable by other legal AI/NLP researchers or not worthy of publication, or that it  cannot make an impact in the legal AI/NLP community, let alone the fact that we have provided evidence that our framework can be applied to produce corpora involving other jurisdictions.
> >
> > [1] Joel Niklaus, Veton Matoshi, Matthias Stürmer, Ilias Chalkidis, Daniel E. Ho. MultiLegalPile: A 689GB Multilingual Legal Corpus. ACL (1) 2024: 15077-15094
> >
> >
> > [2] Ilias Chalkidis, Nicolas Garneau, Catalina Goanta, Daniel Martin Katz, Anders Søgaard. LeXFiles and LegalLAMA: Facilitating English Multinational Legal Language Model Development. ACL (1) 2023: 15513-15535
> >
> > [3] Chaojun Xiao, Haoxi Zhong, Zhipeng Guo, Cunchao Tu, Zhiyuan Liu, Maosong Sun, Yansong Feng, Xianpei Han, Zhen Hu, Heng Wang, Jianfeng Xu. CAIL2018: A Large-Scale Legal Dataset for Judgment Prediction. CoRR abs/1807.02478 (2018)
> >
> > [4] Ilias Chalkidis, Ion Androutsopoulos, Nikolaos Aletras. Neural Legal Judgment Prediction in English. ACL (1) 2019: 4317-4323
> >
> > [5] https://huggingface.co/datasets/AUEB-NLP/ecthr_cases

---

> > ### Author Response · Authors · 2025-08-08
> > **Serious misunderstandings (2/2)**
> >
> > **2. Limited evaluation scope**
> >
> > We provided in our response the results of ROME, a SOTA knowledge update method, into Lawformer and NeurJudge to evaluate its effectiveness. These are adaptable versions of state-of-the-art systems.
> >
> > It seems like the phrase "adaptable versions of recent state-of-the-art models" was taken from the review written by Reviewer 7e8S, who was already satisfied with the results we provided about ROME in our response and has no concerns about broader evaluation. As far as we know, all we need to do is to copy these results into the final version of the paper, so we are not sure what you mean by "may benefit from further attention in a future revision", since this is something we have already addressed in our response and it's no longer the case that multiple reviewers have concerns about broader evaluation. We therefore believe that some misunderstanding must exist here. If you haven't had a chance to read our response, we strongly encourage you to do so.
> >
> > We note that Reviewer 7e8S was asking for adaptable versions of recent sotas. This is a perfectly reasonable request, so we addressed it in our response and they were satisfied. It would, however, not be realistic to expect a new model to be proposed in this submission. A new model that is *meaningful* in this context would be one that would produce not only better results on our dataset, but also state-of-the-art results on existing LJP tasks such as charge prediction and penalty term prediction. Our benchmark is not even about evaluation on these other LJP tasks, so why would it be realistic for us to come up with a new model that can produce (near) state-of-the-art results on these other tasks? It is important to keep in mind that this is the Datasets and Benchmarks track not the Main track, so the focus should be on new datasets and benchmarks, not new models. Models are important as far as providing insights into the proposed dataset/benchmark, and in our case, Reviewer 7e8S is in line with us in that providing adapted versions of the sotas, which we did, is sufficient for this purpose.
> >
> > To sum up,  we hope that our response provided more background about the standard practices in the Legal AI/NLP research community, which in turn can clarify your misunderstanding and help you better appreciate the significance of our work. We understand that properly evaluating our work requires a related background in this area and its usual practices, as well as the ability to view our work in the context of a bigger picture about the field, and it's difficult to appreciate its significance if one simply sees it as just another machine learning benchmark or challenge.

---

> > > ### Comment · Reviewer_y3ZJ · 2025-08-08
> > >
> > > Thank you for the clarifications. I appreciate the additional insights, and I hope the final version can incorporate these points more explicitly.

---

### Decision · Program_Chairs · 2025-09-18

**Decision:**

Accept (poster)

**Comment:**

This benchmark looks at the robustness of LJP across statutory changes in the law.

Strengths

> This represents a realistic roadblock to effective LJP, which so far has been fairly understudied
> The metamorphic testing approach is unique and interesting - the annotated taxonomy of  the 22 type changes Is helpful, and, as the authors discuss in the rebuttal, generalizable across legal jurisdictions.
> The engagement of expert annotators at this scale is admirable and effective at sourcing a meaningful set of reference examples for the benchmark.

Weaknesses
>  There could be more detail to the analysis of model failures (as mentioned by Reviewer y3ZJ ) - it is clear that some models perform well under certain types of changes vs others and it would be interesting to see more engagement on this topic of why, and how other methods (such as the ROME-based methods attempted during rebuttals) might succeed at address the challenges specific to a particular article modification type.

> As raised by Reviewer 7e8S, there’s not clear rationale behind the selection of baselines - though this is a common oversight in ML papers [1], it would still strengthen this work for authors to elaborate on their choice of baselines.

> I find that the other concerns I had (diversity of evaluated models, discussion of Chinese Criminal Code and linguistic scope, etc) are addressed by the author rebuttals.

> LLMs are used in some pre-filtering for a subset of modification types. For example, it is mentioned that LLMs are employed to “edit case facts for revision types T2.1–T2.6, which involve changes to action scopes” and other tasks -- it is incredibly important to be explicit about how exactly LLMs were used in the data creation and curation process and provide details of how human legal experts were used to verify and/or calibrate LLM-based filtering. This is not immediately clear in the current submission.

Asides:

> LJP is just one domain but there’s potential for this to be part of a larger set of benchmark resources to evaluate (1) model robustness to statutory changes and (2) present an interesting benchmark for content temporal distribution shift, similar to a “distribution shift” benchmark such as WILDS [2]. The latter takeaway could make this benchmark interesting for research communities outside the LJP niche.

> It would be interesting to connect the lack of adjustment in statutory shifts to reported model hallucination failures in legal applications. In many audits on LLM hallucination in the legal space [3,4,5], a common failure mode is the lack of adjustment to modified legal settings. This benchmark could provide some perspective in testing for this temporal robustness explicitly (eg. by referencing dated modifications in prompts/ responses, etc).


[1] Liao, Thomas, et al. "Are we learning yet? a meta review of evaluation failures across machine learning." Thirty-fifth Conference on Neural Information Processing Systems Datasets and Benchmarks Track (Round 2). 2021.

[2] Koh, Pang Wei, et al. "Wilds: A benchmark of in-the-wild distribution shifts." International conference on machine learning. PMLR, 2021.

[3] Magesh, Varun, et al. "Hallucination‐Free? Assessing the Reliability of Leading AI Legal Research Tools." Journal of Empirical Legal Studies 22.2 (2025): 216-242.

[4] Dahl, Matthew, et al. "Large legal fictions: Profiling legal hallucinations in large language models." Journal of Legal Analysis 16.1 (2024): 64-93.

[5] Dahl, Matthew, et al. "Hallucinating law: Legal mistakes with large language models are pervasive." Law, regulation, and policy (2024).